# Nigrostriatal dopamine modulates the striatal-amygdala pathway in auditory fear conditioning

Allen P. F. Chen[1,2], Lu Chen[1], Kaiyo W. Shi [1], Eileen Cheng[1,3], Shaoyu Ge [1] & Qiaojie Xiong [1] ✉

The auditory striatum, a sensory portion of the dorsal striatum, plays an essential role in learning and memory. In contrast to its roles and underlying mechanisms in operant conditioning, however, little is known about its contribution to classical auditory fear conditioning. Here, we reveal the function of the auditory striatum in auditory-conditioned fear memory. We find that optogenetically inhibiting auditory striatal neurons impairs fear memory formation, which is mediated through the striatal-amygdala pathway. Using calcium imaging in behaving mice, we find that auditory striatal neuronal responses to conditioned tones potentiate across memory acquisition and expression. Furthermore, nigrostriatal dopaminergic projections plays an important role in modulating conditioning-induced striatal potentiation. Together, these findings demonstrate the existence of a nigro-striatal-amygdala circuit for conditioned fear memory formation and expression.

The neural mechanisms through which sensory information is transformed into behavior through learning are fundamental to neuroscience. Accumulating evidence shows that the basal ganglia is an important neural center for learning and memory[1–5]. The tail of the dorsal striatum, also called the auditory striatum, is a substrate for associations between an auditory cue and reward-seeking behavior[6]. After establishing cue-behavior associations, neural activity in the auditory striatum and its cortical, thalamic, and dopaminergic inputs are important for supporting ongoing auditory discrimination behavior in operant tasks[7–9]. It is unknown, however, whether dorsal striatal activity is also employed and required for learning of aversive behaviors.

Much of our understanding of aversive behavior and associative learning comes from classical fear conditioning paradigms[10,11]. The basic ability to form tone-shock associations has classically focused on the amygdala and involves the recruitment of both limbic and sensory neural systems[12–17]. Prior evidence suggests that coincident sensory input to the amygdala from the medial geniculate body and limbic input from the periaqueductal gray area facilitates plasticity within the amygdala for the formation of tone-shock associations[10,18,19]. Furthermore, the orchestration of multiple interactions among thalamic nuclei, amygdala subnuclei, and resident interneurons controls both the formation and behavioral expression of tone-shock associations[13,20–22]. Despite the role of the different striatal regions in operant learning and the finding that ventral striatum receives negative valence from basolateral amygdala to drive negative reinforcement[23], there is little evidence for how the dorsal striatum may integrate with limbic and amygdalar networks to facilitate aversive associative learning.

Traditionally, experimental and theoretical paradigms demonstrate that ventral and dorsal regions of the striatum enable learning of reward-cue associations in the reward prediction error framework[24–26]. However, emerging neuroanatomical evidence highlights that different regions of the striatum have different inputs and outputs, suggesting different behavioral functions[27,28]. Recent work suggests that the auditory striatum and its dopaminergic inputs facilitate behavioral responses to threats[5,29,30]. These observations highlight the auditory striatum as a functionally distinct neural area that may underlie the

[1]Department of Neurobiology and Behavior, SUNY Stony Brook, Stony Brook, NY 11794, USA. [2]Medical Scientist Training Program, Renaissance School of Medicine at SUNY Stony Brook, Stony Brook, NY 11794, USA. [3]Department of Physiology and Biophysics, SUNY Stony Brook, Stony Brook, NY 11794, USA. ✉e-mail: qiaojie.xiong@stonybrook.edu

learning of fear behaviors. Is striatal plasticity important for the formation of associations between cues and aversive outcomes? How does the auditory striatum functionally connect with downstream circuits to enable sensory- or threat-based behaviors?

Here, we examined whether the auditory striatum integrates with fear-associative circuits and displays plasticity during auditory fear conditioning. Furthermore, we investigated possible modulatory sources of striatal plasticity in fear learning. Using optogenetic inhibition, we found that the auditory striatum is necessary for classical auditory fear conditioning in mice. We also identified the lateral amygdala as a downstream circuit to which the auditory striatum relays information about aversive cues, as ablation of amygdala-projecting auditory striatal neurons impaired auditory fear responses. Furthermore, using deep brain microendoscopic imaging in behaving mice, we found that both striatal neuronal and dopamine responses to tones were markedly potentiated during fear conditioning. Inhibiting auditory striatal dopaminergic activity impaired striatal tone potentiation and behavioral fear responses. Overall, these findings reveal that the auditory striatal circuit is a key locus for tone-induced fear acquisition, which advances our understanding of the role of the basal ganglia in learning and memory.

## Results

### Auditory striatal activity is required for auditory fear conditioning

To explore the role of the auditory striatum in aversive behavior, we first determined whether it is required for auditory fear conditioning using a behavioral paradigm (Fig. 1a) based on previous work[31–33]. Briefly, on the conditioning day, mice were placed into a chamber (Context A) with metal bar flooring. Eight 20-s habituation tones were presented followed by eight 20-s conditioning tones. Each conditioning tone co-terminated with a 2-s foot shock. Twenty-four and 48 h after conditioning, mice underwent two probe sessions in a different context (Context B). During each probe session, sixteen 20-s tones were played without foot shocks. All tones during conditioning and probe sessions were randomly interspersed with 100- to 180-s intertrial intervals.

Using this fear conditioning paradigm, we measured the fear responses of mice when suppressing auditory striatal neuronal activity during conditioning using previously validated optogenetic silencing techniques[7,8]. ArchT[34], an inhibitory opsin, was expressed in auditory striatal neurons via bilateral adeno-associated virus (AAV) infusion into adult mice. In a separate group of mice, green fluorescent protein (GFP) only was expressed as a control. In both groups, optical fibers were implanted above the auditory striatum after viral infusion (Fig. 1b). Three weeks after surgery, mice underwent fear conditioning. Control mice without optogenetic intervention showed typical freezing behavior in response to tones associated with shocks (Fig. 1c). Fear behavior was analyzed by measuring the amount of freezing, which was $7.28 \pm 0.89\%$ during habituation, $40.52 \pm 6.23\%$ during conditioning, $38.72 \pm 3.92\%$ in the first probe session, and $16.23 \pm 5.70\%$ in the second probe session.

To silence auditory striatal neurons during conditioning when the tones were paired with a foot shock, we delivered light stimulation beginning 2 s prior to tone onset and terminating with tone offset (Fig. 1d). In probe sessions, no optogenetic light was delivered, but mice were attached to the patch cords when conditioned tones were delivered. After all sessions, mice were euthanized and analyzed to confirm effective ArchT expression patterns and proper optic fiber implantation sites (Supplementary Fig. 1a). We found that optogenetic inhibition of auditory striatal neuronal activity during tone-shock pairings impaired the freezing of ArchT-infused mice compared with GFP control mice during the probe sessions (Fig. 1e, Supplementary Fig. 1b). We next sought to determine whether auditory striatal activity is necessary for expression of conditioned tone-induced freezing after successful learning. To do this, in a separate group of mice, we inhibited the

auditory striatum during conditioned tone presentations during the first probe session (Fig. 1f). We found that compared with GFP control mice, optogenetic inhibition during conditioned tone presentation decreased freezing time during only the first but not the second probe session (Fig. 1g, Supplementary Fig. 1c). To exclude the possibility that the observed effects were due to an overall disturbance of auditory striatal neuronal activity, we randomly inhibited the auditory striatum the same number and duration of times during intertrial time intervals outside of tone presentations during the probe sessions (Supplementary Fig. 1d). This inhibition regimen did not have a detectable impact on freezing (Supplementary Fig. 1e and f). We also note that outside of the task, optogenetic light delivery did not have a significant impact on gross movement parameters (Supplementary Fig. 1g).

Taken together, using optogenetic techniques to specifically silence auditory striatal neurons, we found that auditory striatal neuron activity is required for both the learning and expression of tone-induced fear memory.

### Auditory striatal neurons projecting to the lateral amygdala are required for auditory fear conditioning

The observed function of the auditory striatum in the formation and expression of tone-shock associations led us to investigate the neural circuits through which the auditory striatum may integrate to enable aversive behavior. Toward this end, we analyzed the projections of auditory striatal neurons by expressing GFP in auditory striatal neurons in mice. We found that the lateral amygdala, a key component of the neural circuit for fear conditioning, receives direct projections from the auditory striatum (Fig. 2a). Injection into this region exhibits little viral spread towards the lateral amygdala, across animals of varying levels of virus diffusion (Supplementary Figure 2a). To confirm the existence of these projections, we infused the retrograde canine adenovirus (CAV) expressing GFP into the lateral amygdala[35,36] and identified GFP-positive neurons in the auditory striatum (Fig. 2b).

To determine the functional role of auditory striatal projections to the lateral amygdala in auditory fear conditioning, we genetically ablated these neurons using a dual viral approach (Fig. 2c)[37]. To validate the genetic ablation approach, we first determined that caspase expression could lead to depletion of auditory striatal neurons (Supplementary Fig. 2b). Briefly, either AAV-tdT or AAV-Caspase was injected to the auditory striatum of two respective cohorts of mice. Consistent with optogenetic inhibition of auditory striatal neurons, this method of genetic ablation induced a freezing behavioral deficit (Supplementary Fig. 2c). Notably, this method of ablation resulted in an 82.1% reduction in neuronal density of the auditory striatum (Supplementary Fig. 2d and e Control, $1.095 \pm 0.02$ vs. Ablation, $0.192 \pm 0.03$ neuron x $10^3$/mm$^3$). We next used caspase to specifically ablate auditory striatal neurons projecting to the lateral amygdala. Using a genetically-restricted approach (Fig. 2c), CAV-Cre was bilaterally injected into the lateral amygdala, and AAV-DIO-Caspase[37] was injected into the auditory striatum to specifically ablate auditory striatal neurons projecting to the lateral amygdala. As a control, we infused AAV-DIO-tdTomato instead of Caspase in another group of mice, which were verified to have either tdTomato expression or a decrease in neuronal density in the auditory striatum (Supplementary Fig. 2f, g). This ablation of auditory striatal neurons projecting to the lateral amygdala resulted in a 32% reduction in neuronal density of the auditory striatum (Control, $1.07 \pm 0.04$ vs. Ablation, $0.72 \pm 0.08$ neuron x $10^3$/mm$^3$). We should also note here that in both ablation strategies, no apparent effects were found in lateral amygdala neuronal density (Supplementary Fig. 2d, f). Using these two groups of mice, we performed auditory fear conditioning. We found a significant decrease in freezing during the probe sessions after ablation of amygdala-projecting auditory striatal neurons compared with control mice (Fig. 2c). This finding suggests that striatal-amygdala projections serve as an indispensable pathway for auditory fear conditioning. However,

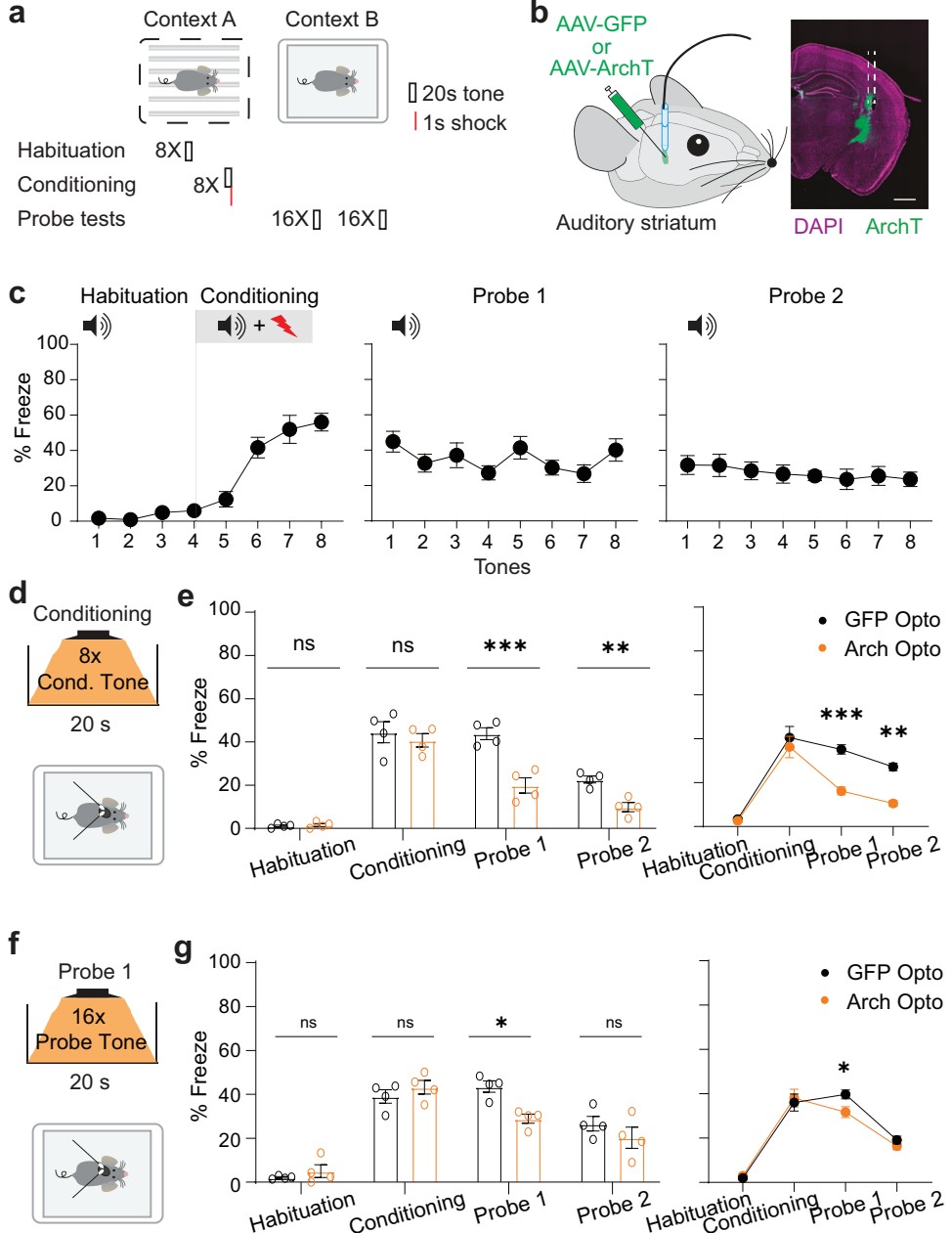

**Fig. 1 | Optogenetic inhibition of the auditory striatum impairs auditory associative fear memory. a** Experimental schematics of auditory fear conditioning. **b** Left, schematic for optogenetic surgery in mice. Bilateral implantation and infusion of either AAV-eGFP or AAV-ArchT were performed. Right, representative immunohistological image. Purple, DAPI; green, Arch-eGFP. Scale bar = 0.5 mm. **c** Freezing in response to tones during habituation, conditioning, and probe sessions. Freezing percentages for two consecutive tones were averaged as a single data point. Error bars are standard error of the mean (SEM; $n = 4$ mice). **d** Schematic for inhibition during conditioned tone presentation. **e** Left, bar plot of averaged freezing percentage in response to tones during habituation, conditioning, and probe sessions. Individual dots are averaged freezing percentages corresponding to individual animals. Right, line plot of averaged freezing percentage in response to tones during habituation, conditioning, and probe sessions. Error bars are SEM ($n = 4$ mice per group; two-sided unpaired Mann-Whitney test; ns, $p > 0.05$; **$p = 0.0028$; ***$p = 0.0007$). **f** Schematic for inhibition during tone presentation in the first probe session. **g** Left, bar plot of averaged freezing percentage in response to tones during habituation, conditioning, and probe sessions. Individual dots are averaged freezing percentages corresponding to individual animals. Right, line plot of freezing percentage in response to tones during habituation, conditioning, and probe sessions. Error bars are SEM ($n = 4$ mice per group; two-sided unpaired Mann-Whitney test, ns, $p > 0.05$; *$p = 0.0286$). Source data are provided as a Source Data file.

which population of amygdala neurons receive striatal projections and regulate amygdalar circuits remains elusive.

### Auditory striatal neuronal responses to tones are potentiated during auditory fear conditioning

We next asked whether the auditory striatum responds to tones differently after they are paired with foot shock during conditioning, by

which it might actively regulate auditory fear memory. To answer this question, we recorded neuronal activity in the auditory striatum via in vivo $Ca^{2+}$ imaging in behaving mice. We infused AAV9-CAG-GCaMP6f into the auditory striatum and implanted a GRIN lens above the infusion site[7,38] (Fig. 3a). We allowed viral incubation and expression to occur for at least 3 weeks before baseplate implantation. Seven days after baseplate implantation, mice were habituated to camera

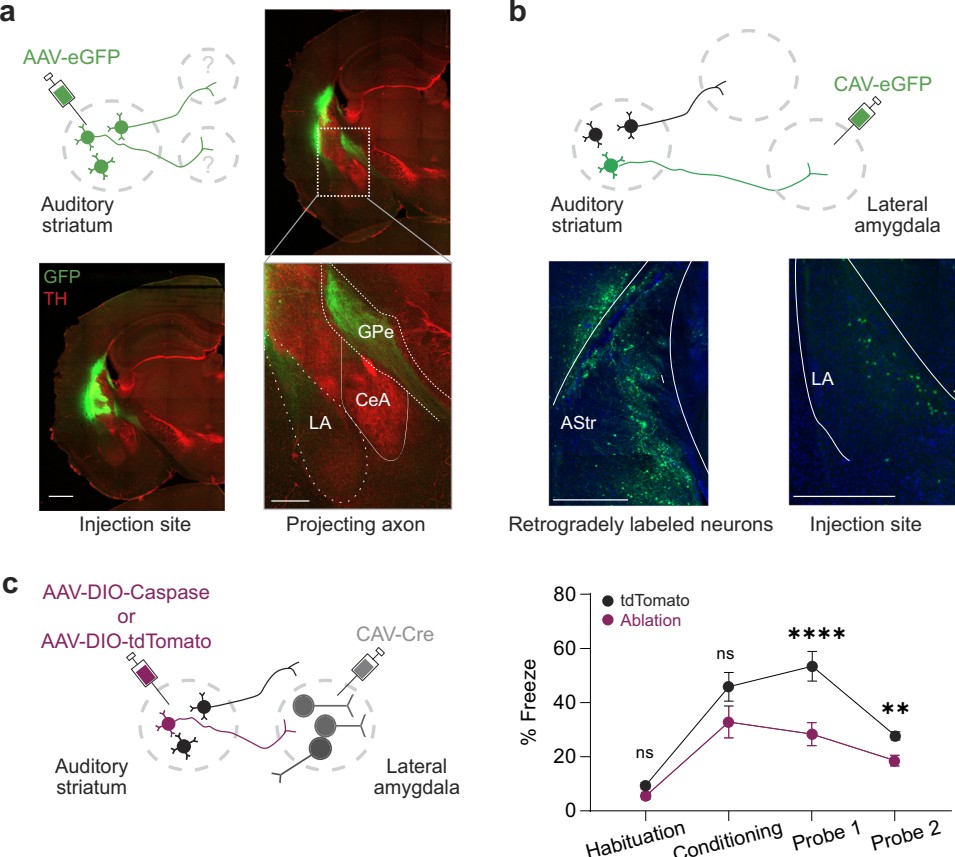

**Fig. 2 | Functional projections from the auditory striatum to the lateral amygdala are necessary for fear memory formation. a** Top left, schematic for anterograde tracing from the auditory striatum to the lateral amygdala using eGFP. Bottom left, histological image of injection site viral spread in the auditory striatum. Top and bottom right, histological images of auditory striatal axons in the lateral amygdala at 10 × and 20 × magnification, respectively. Scale bars = 1.0 mm for injection site and projection images and 0.20 mm for the 20 × image. LA, lateral amygdala; CeA, central amygdala; GP, globus pallidus. **b** Retrograde tracing from the lateral amygdala was performed by infusing CAV-eGFP into the lateral amygdala (right image). Retrograde labeling occurred in the auditory striatum (left image). Blue, DAPI; green, eGFP. AStr, auditory striatum; LA, lateral amygdala. Scale bars = 0.25 mm. **c** Left, schematic for retrograde ablation of auditory striatal neurons projecting to the lateral amygdala using CAV-Cre and AAV-DIO-Caspase. Right, averaged freezing percentage across animal cohorts in response to tones during habituation, conditioning, and probe sessions. Error bars are SEM ($n = 4$ mice for control, $n = 4$ mice for ablation; two-sided unpaired Mann-Whitney test, ns, $p = 0.098$; **$p = 0.0027$; ****$p = 0.000018$). Source data are provided as a Source Data file.

mounting for 4 days before auditory fear conditioning. We recorded GCaMP6f signals from auditory striatal neurons during each task session. In collected images, regions of interest (ROI) were detected by CNMF-E and registered using the CellReg function[7,39], which facilitated the recording of auditory striatal neurons throughout habituation, conditioning, and probe sessions (Fig. 3b). Recordings from each neuronal ROI were aligned to the onsets of conditioned tones and sorted according to the magnitude of tonal response within a 1 s time window. The peak Z-score, representing the fluorescence change in response to the tones, was measured and plotted for each task session (Fig. 3c, top panels). The number of neurons with increased, decreased, or unchanged activity in response to the tones was quantified (with magnitudes of changes compared with baseline using Wilcoxon rank-sum tests; Fig. 3c bottom panels, Supplementary Fig. 4a). We found a substantial increase in the neuronal population with increased responses to conditioned tones during the conditioning and first probe sessions (Fig. 3d). Interestingly, we observed a gradual and significant increase in tonal responses during the conditioning session across the eight conditioning tones (Fig. 3e); of note, tone-responsive neurons do not substantially overlap with foot shock-responsive neurons (Supplementary Fig. 4b), suggesting a modulation of striatal tone response activity after consecutive foot shock pairings.

To exclude the possibility that the striatal tonal potentiation was an overall aversive state-based modulation[40], we investigated whether this phenomenon occurred toward innocuous unpaired tones. To address this, we analyzed mice subjected to non-paired tones before and after conditioning. In this paradigm, mice were subjected to two different sets of tones but only conditioned to one set of tones (Supplementary Fig. 3a). Mice were habituated to 10-kHz tones (Tone B) and subsequently conditioned to 5-kHz tones (Tone A). Striatal tonal responses were observed toward both tones in the pre-conditioning session (Supplementary Fig. 3b). After tone-shock pairings with Tone A, responses to Tone A but not Tone B potentiated (Supplementary Fig. 3b, c). Furthermore, animals did not exhibit behavioral freezing toward Tone B as they have towards the conditioned Tone A (Supplementary Fig. 3d). This suggests that striatal tonal potentiation was specific to conditioned tones. All mice used in these experiments were confirmed to have proper lens placement within the auditory striatum as well as GCaMP6f expression (Supplementary Fig. 3e).

Although there were overall increased freezing behaviors after conditioning, mice exhibited a distribution of freezing when we calculated individual freezing time towards each tone presentation (Supplementary Fig. 3f–h). To explore whether neuronal tonal responses are correlated with freezing levels, we calculated individual

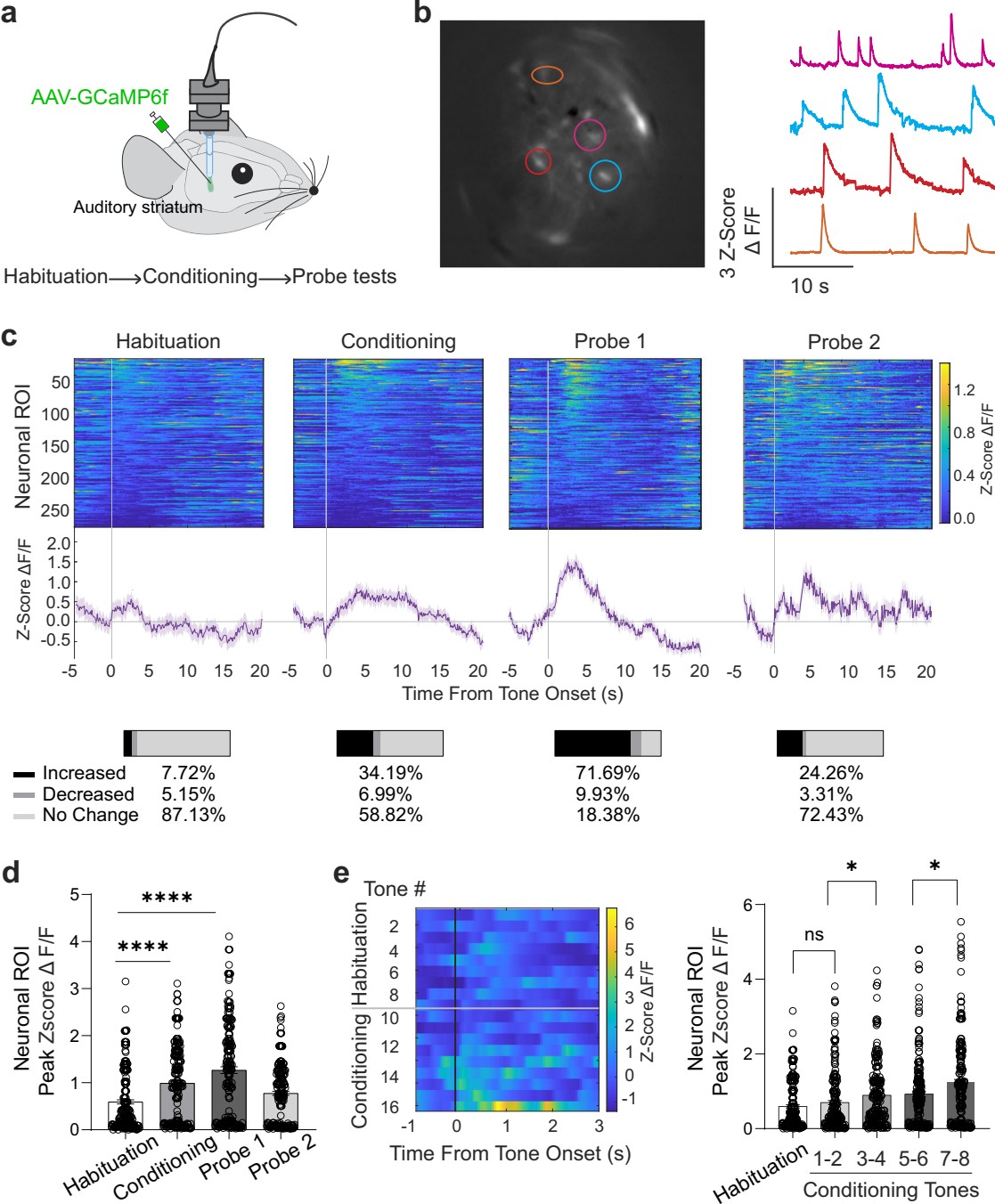

**Fig. 3 | Auditory fear conditioning potentiates conditioned tone-induced activity in the auditory striatum. a** Schematic for simultaneous microendoscopic imaging of auditory striatal Ca²⁺ activity. **b** Sample image and corresponding Ca²⁺ events. **c** Upper row, heatmaps of neuronal ROI responses ($n = 262$ neurons from 6 mice) to tones during habituation, conditioning, and probe sessions. Middle row, averaged neuronal ROI responses for the corresponding sessions. Bottom row, quantification of the proportion of neurons with increased (black), decreased (dark gray), or no significant change (light gray; two-sided Wilcoxon rank-sum test, $p = 0.0434$) in response to tones. Error bars in traces are SEM. **d** averaged neuronal activity in response to tones during habituation, conditioning, and probe sessions. Individual dots are single neuronal ROI averaged responses towards tone

presentation. Error bars are SEM ($n = 211$ averaged neuronal ROI tonal responses, extracted across 6 mice; two-sided Wilcoxon rank-sum test, for habituation vs. conditioning, ****$p = 0.000029$; for habituation vs. probe 1, ****$p = 0.000018$). **e** Left, heatmap of neuronal ROI responses to tones during a habituation-conditioning session. Right, averaged neuronal activity in response to tones during habituation tones and conditioning sessions. Responses to two sequential tones during conditioning were binned together. Individual dots are single neuronal ROI averaged responses towards tone presentation. Error bars are SEM ($n = 211$ averaged neuronal ROI tonal responses, extracted across 6 mice; two-sided Wilcoxon rank-sum test; ns, $p = 0.078$; *$p = 0.0211$; *$p = 0.0398$). Source data are provided as a Source Data file.

neuronal responses to tones (averaged peak Zscore ΔF/F from all responsive probe sessions), and plotted them against the corresponding freezing percentages in the same sessions. Using Pearson correlation, we find that there is a moderate positive correlation

between the tonal responses and freezing levels (Supplementary Fig. 3i, $R = 0.3105$; $p = 1.88 \times 10^{-5}$).

Altogether, we found that auditory striatal responses specifically to conditioned tones were potentiated during tone-foot shock pairings

and this potentiation continued through the first probe session. These findings suggest a regulatory role of the auditory striatum in auditory fear conditioning.

## Elevated dopamine in the auditory striatum during auditory fear conditioning

The potentiation of auditory striatal neuronal activity during fear conditioning led us to investigate the underlying regulatory mechanisms. Several recent studies, including one from our group, report that dopaminergic neurons are activated in operant tasks and regulate task-related decision-making[4,7,41]. Emerging evidence also suggests that dopaminergic activity is triggered by and possibly regulates cue-associative aversive behavior[5,33,42]. We therefore examined dopaminergic activity in the auditory striatum and its possible role in auditory fear conditioning.

We first combined a CAV-Cre retrograde viral approach with in vivo Ca²⁺ imaging to specifically label auditory striatal-projecting substantia nigra pars compacta (SNc) neurons with GCaMP6f (Fig. 4a)[7].

To validate this approach, we used fluorescent protein-based tracing to determine whether injection of CAV virus in the auditory striatum could label the dopaminergic neuronal population projecting to the auditory striatum (Supplementary Fig. 5a, b). Using this strategy, we find that most retrogradely labeled neurons are within the midbrain dopamine region, the SNc with a minor population within the ventral tegmental area (Supplementary Fig. 5b: SNc, $94.4 \pm 1.14\%$ vs. VTA, $5.6 \pm 1.14\%$; Mean ± SEM). This is consistent with prior results demonstrating the non-overlapping and specific dopaminergic population projection to the auditory striatum[5,7]. We utilized the same viral labeling strategy to monitor auditory striatal neurons projecting to the auditory striatum[7]. Briefly, we injected CAV-Cre into the auditory striatum and the Cre-dependent expression vector GCaMP6f into the SNc. A GRIN lens was implanted above the SNc after viral infusion. After 3 weeks of recovery, we performed baseplate implantation to obtain a proper imaging field-of-view. We subsequently subjected mice to auditory fear conditioning during which the GCaMP6f signal was captured. Mice exhibited conditioned freezing, consistent with our

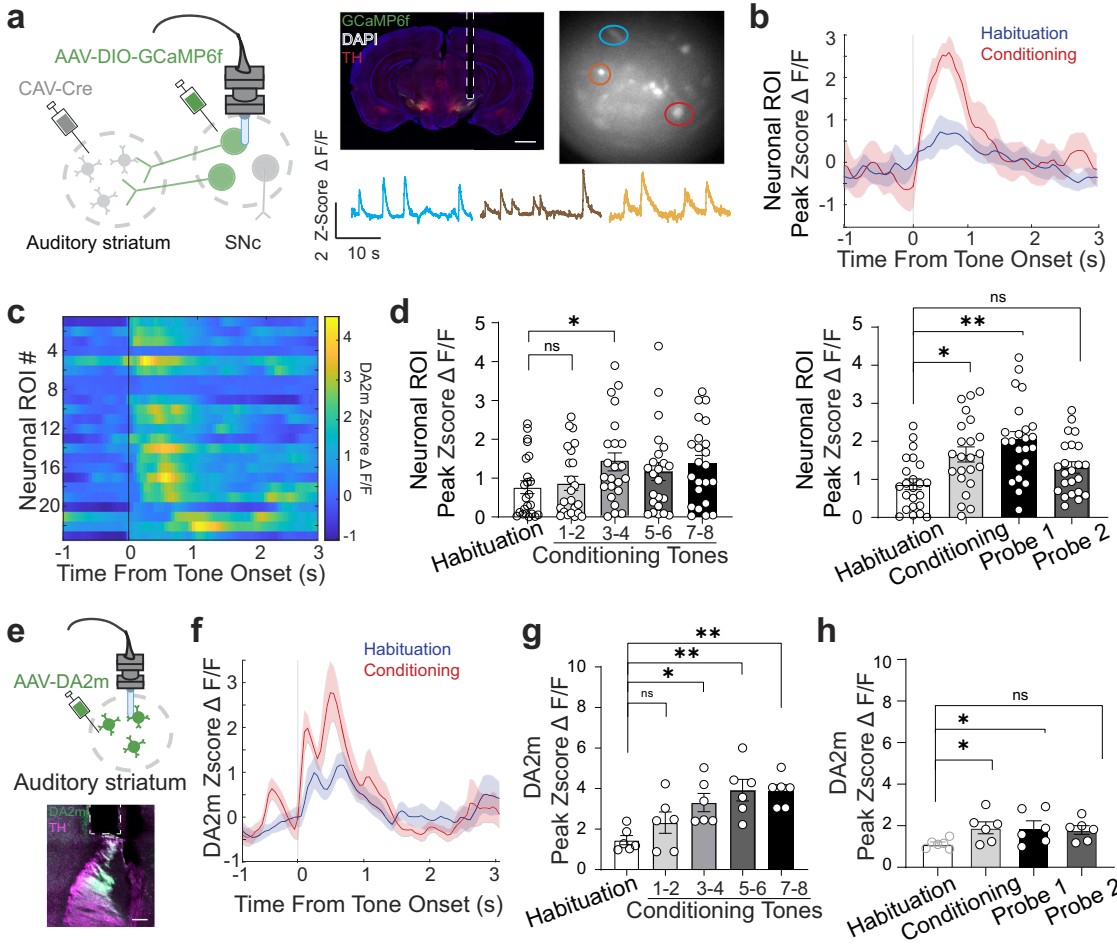

**Fig. 4 | Auditory striatum dopamine responses potentiate during auditory fear memory formation and expression. a** Left, schematic for imaging auditory striatum projecting SNc neurons. Left middle, representative image of lens implantation above the SNc. Red, tyrosine hydroxylase marker; green, GCaMP6f; blue, DAPI. Scale bar = 1 mm. Right middle, representative field-of-view image of SNc. Right, representative traces of neuronal ROI activity. Scale bars = Z-score ΔF/F and 10 s. **b** Representative SNc neuronal response to tones in habituation (blue) and conditioning (red) sessions. **c** Averaged tonal response latencies for individual SNc neuronal ROI (n = 23 from 6 mice). **d** Averaged neuronal tonal responses in habituation, conditioning, and probe sessions. Responses to two sequential tones during conditioning were binned together in the left graph. Each dot represents an SNc neuronal ROI (two-sided Wilcoxon rank-sum test; ns, p = 0.9231, *p = 0.0201,

**p = 0.0082; n = 23 neuronal ROIs from 58 detected, n = 6 mice). Error bars are SEM. **e** Top, schematic for microendoscopic dopamine sensor imaging. Bottom, representative histological image of lens implant site and DA2m expression in the auditory striatum. Green, DA2m; purple, TH. Scale bar = 200 μm. Error bars are SEM. **f** Representative traces of auditory striatal dopamine tonal responses in habituation (blue) and conditioning (red) sessions. Error bars are SEM (n = 8 traces per condition). **g, h** Auditory striatum dopamine tonal responses in habituation, conditioning, and probe sessions. Responses to two sequential tones during conditioning were binned together in (**g**). Error bars are SEM; each dot represents one mouse FOV (Wilcoxon rank-sum test; ns for habituation vs. conditioning tones 1-2, p = 0.38; ns for habituation vs. probe 2, p = 0.12 **p = 0.003; *p = 0.019; n = 6 mice). Source data are provided as a Source Data file.

earlier experiments (Supplementary Fig. 5c). When we aligned the activity of auditory striatum-projecting SNc neurons with tone onset (Fig. 4b&c), we found that the tonal responses increased in magnitude throughout tone-shock pairings (Fig. 4d left graph). Such tonal responses appeared to continue through the first probe session (Fig. 4d right graph). Neuronal ROI's plotted in this analysis were found to have a statistically significant tone response during the behavioral sessions. Mice used in this experiment were verified to have proper lens placement above the SNc and expression of GCaMP6f in dopaminergic neurons (Supplementary Fig. 5c). These results indicate that auditory striatal-projecting SNc neurons were activated during auditory fear conditioning.

We next examined dopamine levels in the auditory striatum using a fluorescence sensor for dopamine, DA2m[43], as performed previously[7] (Fig. 4e). Given the membranous and ubiquitous expression of the sensor, fluctuations would illuminate the entire GRIN lens field-of-view. Therefore, in our analyses, we conducted whole field-of-view fluorescence quantification. We note here that our methodology and analysis is functionally similar to prior use of fiber photometry to capture DA2m signals[43]. We found that dopamine levels rose when fluorescence intensity was aligned with tone onset during habituation and was substantially potentiated during conditioning (Fig. 4f), consistent with our observation of elevated dopaminergic neuronal activity in the SNc. Similarly, during the conditioning session, dopamine levels quickly potentiated across tone-shock pairings (Fig. 4g), and this potentiation continued through the first probe session (Fig. 4h). Only mice with proper lens implantation and expression of DA2m near the site of imaging were included in the analyses (Supplementary Fig. 5d). Using a similar control experiment as in Supplementary Figure 3e, we find that dopamine sensor potentiation is tone-specific and does not potentiate towards unpaired tones (Supplementary Fig. 5e). These results indicate that tone-shock association training enhances dopaminergic signaling in the auditory striatum.

### Inhibition of dopaminergic activation in the auditory striatum impairs auditory fear conditioning

We next determined whether dopaminergic activity during fear conditioning is necessary for the function of the auditory striatum in auditory fear conditioning. We employed an optogenetic approach to inhibit SNc terminal transmission in the auditory striatum. We bilaterally infused AAV-DIO-ArchT into the SNc of DAT-Cre mice followed by implantation of an optic fiber above the auditory striatum (Fig. 5a). Five weeks after surgery, we subjected mice to an optogenetic light inhibition protocol (Fig. 5b) based on our earlier observation of tone-evoked dopamine release in the auditory striatum. After terminal inhibition during conditioned tone presentation in the conditioning session, we found a decrease in freezing in both probe sessions (Fig. 5c). All mice used in these experiments were confirmed to have proper viral expression and implant placement (Supplementary Fig. 6a).

To test whether the decreased freezing upon inhibition of dopaminergic projections was a result of an absence of potentiation of striatal neuronal activity during auditory fear conditioning, we simultaneously measured striatal neuronal activity and nigrostriatal input inhibition (Fig. 5d). To be compatible with in vivo $Ca^{2+}$ imaging in the auditory striatum, we used a chemogenetic approach for nigrostriatal input inhibition. We infused the $Ca^{2+}$ indicator GCaMP6f and CAV-Cre into the auditory striatum and the Cre-dependent chemogenetic silencer hM4Di into the SNc of wild-type mice. Control mice underwent the same surgery but were infused with the fluorophore mCherry with no chemogenetic construct. Five weeks later, we performed baseplate implantation and behavioral handling. During behavioral handling, mice were habituated to intraperitoneal (i.p.) injections of saline. We performed i.p. injections of CNO 30 min prior to auditory fear conditioning. On subsequent probe days, mice did not receive i.p.

injections. Consistent with optogenetic inhibition of the nigrostriatal pathway (Fig. 5a–c), chemogenetic inhibition of striatal-projecting SNc neurons decreased freezing behavior exhibited on probe sessions (Fig. 5g). Thus, using this method, inhibition of nigrostriatal transmission to the auditory striatum impaired fear learning. We next sought to determine how this impacted striatal plasticity dynamics. We observed that, compared with fluorophore-only controls, CNO-mediated chemogenetic inhibition abolished fear conditioning-induced striatal potentiation (Fig. 5e, f). These findings suggest that dopamine activation is required for the potentiation of striatal neuronal activity during auditory fear conditioning.

Altogether, these findings show that projections from the SNc to the auditory striatum are activated and potentiated during auditory fear conditioning, and this potentiation is indispensable for the regulation of auditory striatal neuronal activity during the establishment of tone-shock associations.

## Discussion

In this study, using a tone-shock conditioning paradigm that has classically been used to associate initially neutral auditory tones with aversive events, we discovered that, akin to other regions of the brain that regulate fear memory, the auditory striatum showed potentiated responses during auditory fear conditioning. Using genetic ablation and optogenetic inhibition, we revealed that this striatal activity potentiation was functionally important for fear memory acquisition, likely through the projection to the lateral amygdala. Additionally, we found that striatal dopamine may be a source of plasticity for this potentiation. Striatal dopamine responses were enhanced by fear conditioning and optogenetic inhibition of dopamine responses in the auditory striatum impaired memory acquisition. Together, these findings advance our understanding of how auditory striatal neural circuits contribute to auditory learning and memory through their presynaptic and postsynaptic partners.

An interesting observation is that while we observed enhancement of striatal neural activity as well as dopaminergic transmission during the conditioning, inhibition of these circuits did not appear to impact the development of freezing behavior in conditioning sessions, but to only affect freezing responses in probe sessions. One possible explanation is that the expression of US-associated freezing behavior during conditioning could be primarily dependent on somatosensory circuitry[40,44], with the footshock being a stimulus that is acute and strong enough to mask the additional impact from the auditory circuitry. It is also conceivable that in the context of the behavior, the amygdala could employ other parallel pathways to enforce freezing behavior acutely[18,45]. Along with this, the potentiation of striatal neuronal activity during conditioning may be the plasticity involved in modulation of amygdala over a longer time scale, akin to memory consolidation.

Compared to the more traditional striatal targets such as the globus pallidus, the amygdala receives much less attention. In this study we demonstrated a direct connection from the auditory striatum to the lateral amygdala (Fig. 2 and Supple. Fig. 2). Ablation of amygdala-projecting neurons in the auditory striatum dramatically impaired fear memory. However, our results did not rule out possible participation of collaterals from these neurons to other brain regions. Optogenetic manipulations of striatal terminals in the lateral amygdala would confirm this pathway's specificity. Furthermore, our results do not differentiate the activities of striatal neuronal populations (e.g. D1 and D2 MSNs). Anterograde viral tracing suggests that both D1 and D2 MSNs project to the lateral amygdala. How might dopaminergic modulation through these two pathways impact auditory fear conditioning? Systemic pharmacological antagonism of both D1 and D2 receptors has been shown to abrogate different forms of fear conditioning with mixed interpretations[46–52]. Differential roles for the D1 and D2 MSN pathways within the dorsal and ventral striatal

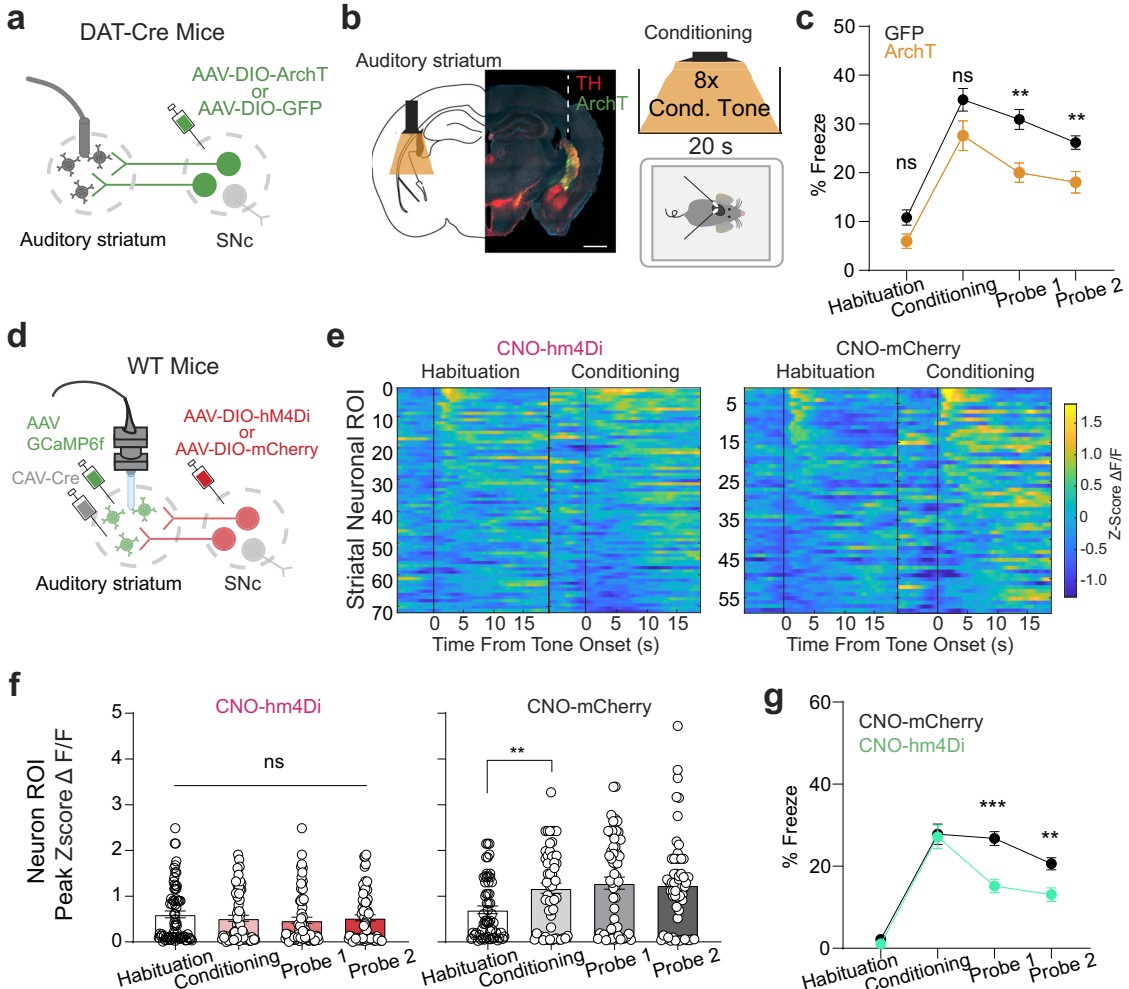

**Fig. 5 | Inhibition of nigrostriatal dopamine dampens striatal potentiation and acquisition of auditory fear responses. a** Schematic for optogenetic surgery in DAT-Cre mice. AAV-DIO-ArchT was bilaterally infused into the SNc, and optic fibers were bilaterally implanted above the auditory striatum. **b** Left, representative immunohistological image. Green, ArchT-eGFP construct expressed in SNc terminals; red, tyrosine-hydroxylase marker; blue, DAPI. Scale bar = 1.0 mm. Right, schematic for optoinhibition light delivery during conditioning session. In subsequent probe sessions, mice were attached to the patch cord, but no optogenetic light was delivered. **c** Line plots of averaged freezing percentage in response to tones during habituation, conditioning, and probe sessions. $n = 4$ mice from each group. Error bars are SEM (two-sided unpaired Mann-Whitney test; ns for habituation, $p = 0.12$; ns for conditioning, $p = 0.08$; for Probe 1 **$p = 0.002$; for Probe 2

**$p = 0.009$). **d** Schematic for chemogenetic inhibition of SNc. **e** Heatmap of striatal neuronal activity aligned to tone onset during habituation and conditioning sessions for mice undergoing CNO-mediated nigrostriatal chemogenetic inhibition. **f, g** Quantification of neuronal peak tone responses (**f**) and averaged tone-induced freezing percentage (**g**) in all sessions from hM4Di and mCherry mice. Individual dots are single neuronal ROI averaged responses towards tone presentation. Error bars are SEM, $n = 5$ mice from each group. For F, a two-sided Wilcoxon rank-sum test was performed (for habituation vs. conditioning ns, $p = 0.34$; for habituation vs. probe 1 ns, $p = 0.49$; for habituation vs. probe 2 ns, $p = 0.39$). For G, a two-sided unpaired Mann-Whitney test was performed (for probe 1, ***$p = 0.0002$; for probe 2, **$p = 0.0078$). Source data are provided as a Source Data file.

subdivisions have been suggested; the pathways may respectively correspond to appetitive-positive-actions or aversive-negative-inhibitions depending on the behavior being studied[53–55]. However, in the context of learning, this D1 and D2 opponency is nuanced, with evidence suggesting that the D1 pathway supports generalized learning while the D2 pathway could aid in refinement of learning[56,57]. In the auditory striatum, both D1 and D2 MSNs are responsive towards auditory cues[58]. Our previous works indicate that the D1 pathway is important for auditory discrimination and task performance, while the D2 pathway appears to be dispensable[7]. Future studies are needed to examine the potential functional differences of D1 and D2 MSNs in fear conditioning.

The dorsal striatum is conceptualized as a hub for behavioral output, controlling motor regions while receiving sensory, motor, and higher-order cognitive glutamatergic input[28,59–61]. In this context, synaptic updating occurs through neuromodulatory control via

canonical sources of dopamine, such as the SNc[26,62,63]. Prior evidence indicates that the sensory-laden tail striatum may directly or indirectly participate in these behaviors. With both sensory and threat reinforcement being tied into this region's function, striatal plasticity is likely required for ongoing behaviors. However, prior to this study, tail striatal plasticity and its necessity for aversive learning has not been established. We found that the auditory striatum exhibits tone potentiation and memory engram-like activity during fear conditioning[10,11,16]. This indicates that the auditory striatum may function similarly to other regions such as amygdalar or hippocampal structures, which were previously found to be important for encoding tone-shock associations. Consistent with this, we found that the auditory striatum is functionally important for developing a conditioned fear response toward auditory cues. Optical inhibition during only conditioned tones on the conditioning day impaired auditory fear memory expression. Thus, striatal neural circuits activated during

conditioning may be necessary for tone-shock acquisition. Importantly, the continuous involvement of recruited striatal circuits may remain indispensable for auditory fear memory retrieval. We note here that while this study utilized auditory cues, as with prior studies[6-9,29], this tail portion of the striatum receives projections from multiple sensory channels including auditory, visual, and somatosensory cortical inputs[27,28,64]. Thus, while we only tested the learning and neural potentiation of auditory cue's, this region may facilitate the learning of multiple sensory modalities as has been shown for the amygdala itself[65,66]. Related to our study, we primarily tested the necessity of auditory processing in the auditory striatum and auditory striatal-projecting SNc neurons, but did not assess its sufficiency nor the role of shock expression. Indeed, we found that neurons in the auditory striatum and the SNc exhibited shock responsivity (Supplementary Figs. 4b and 5h) and thus this aversive somatosensory processing could also contribute to fear memory and modulation of the amygdala. Somatosensory processing has been implicated in the tail striatum and thus future studies could be conducted to investigate whether there is an integration of auditory and somatosensory cues within these pathways[67,68].

Having previously studied the role of dopamine in regulating striatal activity in the auditory striatum[7], we asked whether dopamine provides an upstream source of plasticity. Using projection-specific imaging, we found that auditory striatal-projecting SNc neurons responded to tones and showed increased tonal responsivity after fear conditioning. Consistently, dopamine sensor imaging demonstrated that dopamine levels in the auditory striatum increased in response to neutral sounds and showed further increases in response to the same tones after conditioning. The potentiation towards tones appears to be conditioning-specific, as we did not observe potentiation towards unpaired tones in control experiments (Supple. Figs. 3c, 5f). However, we note here that for SNc neuronal imaging, we did not perform the Tone B control experiment and therefore there could be individual neurons that potentiate based on context or generalized threat processing.

We note here, however, that our CAV-based tracing strategy and labeling of the auditory-striatal projecting SNc neurons is not experimentally designed to specifically label dopaminergic input to the auditory striatum. While we have quantified that the majority of neurons labeled using this strategy are dopaminergic and within the SNc region (Supplementary Fig. 5a, b), a small portion may also be in the VTA and non-dopaminergic. In any case, the SNc neuronal data exhibits a similar trend and observation in comparison to the DA sensor data, suggesting that SNc neuronal dynamics are correlated with downstream dopaminergic release in the auditory striatum.

In addition, our analyses showed that the tonal response latencies from SNc neuronal ROIs are not significantly different from those of striatal DA sensor recording, but significantly shorter than those of striatal neuronal ROI (Supple. Fig. 5g). The kinetics and peak of fluorescence between GCaMP6f (SNc neuronal activity) and DA2m (DA activity) are different and the onset of rise in fluorescence may not be a sufficient quantitative indicator of activity onset (i.e., real time calcium fluctuation may be sufficient to trigger DA release and such a change may not be reflected by in vivo imaging). Thus, from our results it is not clear whether or not striatal DA activity is faster than the SNc somatic activity. Here, it is important to consider the possibility that DA activity could be locally modulated through cholinergic mechanisms, a property yet to be studied in this region[69]. Moreover, intersectional genetics studies have demonstrated that the SNc DA neurons projecting to the auditory striatum is VGLUT2+[70]. This unique co-expression may provide an excitatory or auto-regulatory mechanism that could further explain a dissociation between somatic activity and downstream DA release. Furthermore, a few of the SNc neurons presented in this study have substantially slower dynamics than that of the striatal DA sensor recordings, which

may reflect the heterogeneity of SNc neuronal functions in this behavior.

The lateral amygdala is considered a primary sensory input to the amygdalar network and shares a variety of properties with the auditory striatum. For instance, the lateral amygdala receives convergent input from the auditory cortex, auditory thalamus, and somatosensory cortex[71-73]. The auditory striatum similarly receives convergent input from these sensory regions, and its function appears to be modulated by the medial geniculate body and auditory cortical inputs[6,8,9,29]. Thus, one interpretation of the observed tonal potentiation is that the auditory striatum receives a similar behavioral convergence of neural activity as the lateral amygdala. Such a convergence of neural drive can confer striatal plasticity in a similar manner as the lateral amygdala in fear conditioning contexts[74,75], and this convergence may further propagate to the lateral amygdala to regulate fear behavior. This is an intriguing possibility, as it suggests that upstream regulatory mechanisms of the auditory striatum represent a parallel sensory processing pathway to the amygdala. On a circuit mechanism level, the auditory striatum like other striatal regions, is comprised of solely GABAergic neuronal populations, and thus may provide a disinhibitory drive to gate learning in principle amygdalar neurons, akin to local interneuron population function[13,21,22,76-78]. In this context, evidence indicates that the parvalbumin and somatostatin neuronal populations of the lateral amygdala and basolateral amygdala tightly controls the output and plasticity of the amygdala for fear acquisition, discrimination, as well as for extinction learning (i.e. for fear extinction, the gradual interneuron inhibition of amygdalar output as a mechanism for fear memory extinction and dampening)[76-81]. One possibility is that the auditory striatum could provide inhibition to these local interneurons to disinhibit and enhance lateral amygdala principal neuron activity—thus the auditory striatum may serve as a parallel reinforcement circuit to fine tune the sensory learning functions of the lateral amygdala. Consistent with this, in the lateral amygdala, PV neuronal activity has been found to be decreased throughout fear memory formation[78]. The auditory striatum itself has been found to be involved in operant and reward-based learning and could provide a source of disinhibitory plasticity in a variety of contexts beyond fear learning[6,8,13,22,82,83]. Thus, future work will be required to investigate the different possible valences that the auditory striatum could relay to the amygdala. Furthermore, it will be interesting and important to determine whether auditory striatal input impacts the plasticity of either auditory cortical or auditory thalamic inputs to the lateral amygdala, projections found to be critical for auditory fear memory formation[10,73].

Overall, our findings provide insight into the function of the auditory striatum in fear memory acquisition, highlighting striatal potentiation, dopaminergic input, and amygdala network integration.

## Methods
### Animals
All animal procedures were approved by the Stony Brook University Animal Care and Use Committee under the animal protocol number #824397. All animal procedures were further conducted in accordance with U.S. National Institutes of Health standards. C57BL/6 J (The Jackson Laboratory), and DAT-IRES-Cre (The Jackson Laboratory, 006660) mice were used for this study. Both male and female 2–4-month-old mice were used. Mice were housed under similar conditions as previously reported[7,84], with 30−70% humidity, an ambient temperature of 64−97 °F, with free access to food and water, and under a 12-h light/dark cycle conditions All behavioral experiments were conducted during the dark cycle.

### Stereotaxic procedures
For all surgical procedures, mice were first anesthetized with 4% isoflurane gas/oxygen and placed into a stereotaxic rig. Mice received

continuous delivery of 1% isoflurane. All surgeries were performed in a similar manner as previously described[7,8,84]. Briefly, viral injections and implantation of optic fibers and optical GRIN lenses occurred at stereotaxic positions relative to the brain dural surface as measured by the descending object's tip. Coordinates for the auditory striatum were −1.7 mm AP, ±3.30 mm ML, lateral amygdala (−1.9 mm AP, ±3.80 mm ML, and 2.8 mm DV injected at a ±12° angle compared to median, and −2.5 mm DV and for the SNc were −3.10 mm AP, ±1.50 mm ML, and 4.2 mm DV). First, a dental drill bit was employed to open a small craniotomy, leaving the brain surface intact. Next, a custom glass micropipette (tip diameter of 10–15 μm) was slowly inserted into the brain until reaching just past the desired coordinates and then retracted back. Viral solutions were then pressure-infused through a connected Picospritzer II microinjection system (Parker Hannifin Corporation; rate of 100 nl/min). For the experiments, we adjusted injection volumes according to specific brain regions and experimental purposes.

1. For optogenetic and anterograde tracing experiments constrained to the auditory striatum (Figs. 1 and 2a): -250–300 nl of virus was injected across four dorsal-ventral depths of 2.7, 2.6, 2.5, and 2.4 mm.
2. For retrograde targeting experiments constrained to the lateral amygdala (Fig. 3b, c): -250–300 nl of virus was injected at a single angled-depth. For subsequent retrograde cre-dependent caspase experiments (Fig. 3c), -300 nl of virus was infused in the auditory striatum across the dorsal-ventral depths of 2.7, 2.6, 2.5, and 2.4 mm. We re-emphasize here that for lateral amygdala injections, we used an angled approach (±12°) in order to better restrict targeting to the lateral amygdala.
3. For striatal (Figs. 3 and 4e–h) and retrograde targeted SNc (Fig. 4A–D) microendoscopic imaging injections, -500 and 650 nl of fluorophore virus was injected, respectively. For the retrograde targeted SNc experiments (Fig. 4a–d), -400 nl of CAV-Cre virus was injected into the auditory striatum across the dorsal-ventral depths of 2.7, 2.6, 2.5, and 2.4 mm.
4. For optogenetic inhibition of DA terminals in the auditory striatum (Fig. 5a–d), 650 nl of cre-dependent eGFP or ArchT virus was injected into the SNc of DAT-Cre mice.
5. For simultaneous striatal imaging and chemogenetic inhibition (Fig. 5e–g), 500 nl of GCaMP6f virus was injected into the auditory striatum and 650 nl of cre-dependent mCherry or hM4Di was injected into the SNc.

To prevent diffusion, the needles were maintained in position for 5–10 min. Subsequently, the needles were slowly retracted at a rate of -10 μm/s.

For experiments that required implantations, all implantations were performed immediately after viral infusion during the same surgery[7,8,84,85]. The tips of the implantation objects were placed on the brain surface to measure initial stereotaxic coordinates. Optic fibers or GRIN lenses were then slowly lowered to their desired placements. For optogenetic studies, optic fibers were lowered at a rate of -10 μm/s to 400 μm above the optogenetic viral infusion coordinates. Implantations were secured using ultraviolet light (UV)-cured white dental cement (AC Flow-It) and cyanoacrylate. Afterward, dental cement was used to form a cement head cap (Stoelting). For imaging studies, prior to lens implantation, -400 μm brain tissue was aspirated with care to prevent bleeding. Using a custom 3D-printed metal lens holder, the lens was slowly lowered at a rate of -10 μm/s. GRIN lenses were slowly lowered to -200 μm above the imaging viral infusion coordinates. Lenses were then secured to the surrounding skull using UV-cured white dental cement (AC Flow-It), cyanoacrylate, and a dental cap (Stoelting). A cement basin was formed around the lens to facilitate subsequent baseplate implantation. Lenses were protected with silicone sealant as mice were allowed to recover (Kwik-Cast, World

Precision Instruments). Optic fibers were 0.2 mm in diameter coupled to a 6.4 mm-long ceramic ferrule cannula (Thorlabs, US). GRIN lenses were 7.3 mm with 0.6 mm diameter (Inscopix Inc., Palo Alto CA, 1050-002179) or 6.1 mm in length with 0.5 mm diameter (Inscopix Inc., 1050-002182).

## Viruses
The following viruses were used: CAV2-Cre, CAV2-mCherry, CAV2-GFP (PVM, France), AAV2/8-hSyn-DIO-mCherry (Addgene, #50459), AAV2/5-EF1a-DIO-taCasp3-TEVp (Addgene, #45580), AAV2/9-CAG-DIO-ArchT-GFP (UNC vector core), AAV2/9-CAG-EGFP (Addgene, #37825), AAV2/9-CAG-DA2m (WZ Bioscience, US; Yulong Li lab), AAV2/9-CAG-GCaMP6f (Addgene, #100836), AAV2/1-hSyn-DIO-GCaMP6f (Addgene, #100837), and AAV2/8-hSyn-DIO-M4Di-mCherry (Addgene, #44362).

## Auditory fear conditioning
The auditory fear conditioning paradigm consisted of a single habituation-conditioning session followed by two probe sessions 24 and 48 h afterward. Fear conditioning and probe sessions occurred in Context A and Context B, respectively, within a sound-attenuating box (Ashburn, VA). Context A contained an electric grid floor that delivered foot shocks, a lemon scent applied to the chamber (7th Generation, Fresh Scent), and alternating black and white stripes on the walls. Context B contained a white platform and/or transparent cylindrical housing and a mild ethanol scent applied to the chamber. Between sessions, each chamber was cleaned with 70% ethanol solution. All data and videos were collected via an automated video processing system (Freezeframe, Actimetrics). Experimenters were blinded to all ablation and intervention conditions.

During the habituation-conditioning session in Context A, after a short baseline period, eight habituation tones (80 dB tone, 5 kHz) were delivered with randomly generated intertrial intervals (100–180 s). This was followed by eight pairings of the tone with co-terminating foot shocks (0.8 mA). On subsequent probe days, mice received 16 presentations of the conditioned tones without foot shocks with random intertrial intervals (60–100 s) in Context B. The percentage of time spent freezing was calculated using Freezeframe software. Thresholds for freezing behavior were determined by manual inspection of videos. This threshold of quantification distinguishes freezing behavior from generalized pause-groom behavior. Freezing is quantified as a function of averaged CS presentation intervals throughout either conditioning or probe sessions. Overall quantification of freezing percentage results from averaging of freezing percentage at fixed time intervals. Afterward, freezing percentages as a function of equidistant time intervals were plotted and analyzed by MATLAB r2020b and GraphPad software (Mathworks, Natick MA; GraphPad Software Inc., San Diego, CA). Similarly, movement quantification was extracted from Freezeframe software and calculated using MATLAB.

For Tone A/B conditioning experiments (Supplementary Fig. 1), mice underwent a pre-conditioning session in which they were presented with eight 20-s, 10-kHz tones (Tone B). Then, conditioning and probe sessions were conducted with 5-kHz tones (Tone A) as described above. Finally, mice underwent a probe session with 16 20-s, 10-kHz tones (Tone B).

For experiments examining the impact of ablation and optogenetic interventions on general movement, mice were placed in a context distinct from that used in fear conditioning experiments and allowed to freely explore for 25-30 min. Freezing and movement were quantified in the same manner as described above.

## Optogenetic inhibition
Optogenetic experimental procedures were conducted similarly to previous studies[7,8]. For all optogenetic inhibition studies, a bilateral inhibition scheme was employed, and a control fluorophore group of mice underwent the same procedures except for the expression of

optogenetic protein. Specifically, we performed bilateral optoinhibition of auditory striatal activity and bilateral inhibition of dopamine terminal activity. The procedures for these two experimental setups were similar. For the optoinhibition of auditory striatal activity, wild-type C57BL/6 J male and female mice were used. For the optoinhibition of dopaminergic terminals, DAT-Cre (C57BL/6 J background) male and female mice were used. Briefly, fabrication and hand-polishing of optic fibers (0.2 mm in diameter) were performed (Thorlabs, US). The fibers used in these experiments were verified to transmit a 10-mW, 530-nm laser output when connected to a patch cord attached to a solid-state laser (Shanghai Dream Lasers, Shanghai, China). Afterward, simultaneous bilateral viral infusion and implantation were performed. Mice were then allowed to recover for 4–5 weeks to allow for viral expression. One week prior to fear conditioning, mice were acclimatized to handling. On training and experimental days, an FC/PC patch cord (Thor Labs) was attached to the optic cannulae of mice. The solid-state laser was connected to a Master-8 pulse stimulator (A.M.P.I., Israel). Light delivery was controlled by Freezeframe software (Freezeframe, Actimetrics) through the Master 8 pulse generator. Control (eGFP fluorophore expression only) and experimental (ArchT expression in striatal or SNc dopaminergic neurons) mice were habituated to continuous periods of patch cord light delivery (10-20 s) for two sessions in an unrelated context prior to fear conditioning.

For primary experiments on the conditioning day, light was continuously delivered starting 2 s prior to tone onset and for the entire duration of conditioning tones (light offset occurred with tone offset). For subsequent probe sessions, mice were connected to the patch cord in the chamber, but no optogenetic light was delivered. For inhibition during tones paired with foot shocks, light was delivered 2.0 s prior to tone onset until tone offset (22 s continuous light during all tone-shock presentations). For inhibition during tones during probe sessions, light was delivered 2 s prior to tone onset until tone offset (22 s continuous light during all tone presentations). For inhibition randomly interspersed during intertrial intervals, light was delivered continuously for 22 s for 16 repetitions to match the number of tones during conditioning and probe sessions. For inhibition randomly occurring during a 25–30-min session in which mice were allowed to freely move in a separate context, light was delivered continuously for 22 s for 16 repetitions to match the number of tones during conditioning and probe sessions. All intervals were randomly chosen from a Gaussian distribution.

### Chemogenetic inhibition

Simultaneous microendoscopic imaging and chemogenetic inhibition were performed similar to behavioral experiments performed in a prior publication[7]. Briefly, in wild-type C57BL/6 J male and female mice, viral infusion of CAV-Cre to the bilateral auditory striatum, AAV-GCaMP6f to the left auditory striatum, and cre-dependent AAV-hM4Di to the SNc, was performed in one surgery. Simultaneously, lens implantation to the left auditory striatum was performed. Mice were allowed to recover for at least 3 weeks before baseplate implantation. After recovery, mice were then habituated to the experimenter through behavioral handling for 1 day. Mice were mounted with the endoscopic camera and allowed to habituate to the camera during free movement. Subsequently, the animals were habituated to i.p. injections of 300 μl water (vehicle) for 3 days. After habituation towards handling and i.p. injections, one mouse cohort dedicated to mice vehicle and one mouse cohort dedicated to CNO (Enzo; i.p. 2.5 mg/kg, 300 μl diluted in water) were subjected to injections 30 min prior to fear conditioning and simultaneous in vivo imaging sessions.

### In vivo imaging experiments

Microendoscopic surgical and imaging procedures were similar to those used in prior studies[7,38,86]. Briefly, wild-type C57BL/6 J male and female mice underwent simultaneous GCaMP6f or DA2m viral infusion

and GRIN lens implantation. After surgery, mice recovered for 3–4 weeks or until sufficient expression of fluorescent protein was observed by the microendoscopic camera. Baseplate implantation surgery was then performed to fix the camera's field-of-view in place. After baseplate implantation, mice were allowed to recover for 3–7 days. During this time, mice were acclimatized to handling. Microendoscopic cameras were then mounted onto mice, allowing them to freely move and habituate to camera attachment. Mice were habituated to camera attachment for 1–2 days.

Imaging recordings were acquired using the nVista2.0 system (Inscopix Inc.). For all sessions, imaging recordings began prior to Freezeframe session initiation. Recordings were terminated immediately after session conclusion. Using pulses generated by Freezeframe software, signals timed according to tone onset and offset were sent to Inscopix recording software. This timestamped data in conjunction with the recordings allowed for $Ca^{2+}$ and dopamine sensor transient data alignment. A 20-Hz frame rate was used for all recordings. The power intensity of the camera was set at 10–50% LED power with a digital gain of 2-3.

### Image recording, data processing, and analysis

Recording data were processed and analyzed using procedures adapting Inscopix-based hardware, software, and custom written MATLAB code[7,38,86]. Briefly, datasets were preprocessed, spatially downsampled, motion-corrected, and cropped using Inscopix-based Mosaic software (Mosaic, Inscopic). The resultant files were then used for downstream quantification and analysis.

For $Ca^{2+}$ neuronal ROI imaging, we employed the CNMF-E package to determine neuronal ROI activity[87]. We subsequently used the Cell-Reg package to register neuronal ROIs across imaging sessions[39]. Extracted spatial components were manually inspected for proper cellular shape and $Ca^{2+}$ transient dynamics across behavioral sessions. The extracted C_raw component was used to elicit the ΔF/F factor. S, the deconvolved $Ca^{2+}$ signal, was used to register $Ca^{2+}$ event rates. Timestamps recorded using Freezeframe and Inscopix software were used to align the $Ca^{2+}$ transients to tone onset across conditioning and probe sessions. All datasets were analyzed using MATLAB, with further statistical analyses performed using GraphPad Prism 8 (GraphPad Software Inc., San Diego, CA). For dopamine sensor data, a maximal ROI was drawn using the Mosaic software to determine a continuous ΔF/F metric across behavioral sessions. We used the mean fluorescence across each individual session as the F0 component. This ΔF/F value was subsequently used to align with tone onset across conditioning and probe sessions.

### Histology, immunostaining, and confocal image processing

Histology was performed as previously described[7]. Briefly, for all experiments, mice were euthanized using 4% isoflurane through chamber delivery, after which they received an anesthetic dose of urethane (i.p., 250 μl/g). Mice were subsequently transcardially perfused with 4 °C phosphate-buffered saline (PBS) followed by a 4% paraformaldehyde/PBS mixture. Mouse brain tissue was then incubated and fixed in 4% paraformaldehyde/PBS for 16-24 h. Brains were washed in PBS prior to vibratome sectioning (Leica Microsystems). Coronal 50-μm sections were prepared. Immunofluorescence labeling was then performed. After washing in PBS, penetration was performed by incubating sections in 0.50% PBST (PBS + 0.50% Triton-X) for 20-30 min at room temperature, followed by a penetration-blocking step involving incubation of sections in 0.25% PBST with 1% donkey serum for 1 h at room temperature. Sections then were incubated in 0.25% PBST with 1.0% donkey serum with a primary antibody cocktail overnight at 4 °C. Afterward, sections were incubated with 0.25% PBST with a secondary antibody cocktail at room temperature for 3-4 h. Sections were washed with PBS and then mounted using DAPI Fluoromount-G™ (Thermofisher).

The following primary antibodies and concentrations were used: The following primary antibodies and concentrations were used: goat anti-GFP (1:1000, Rockland 600-101-215), chicken anti-GFP (1:1000, Abcam ab13970), goat anti-RFP (1:1000, Rockland 200-101-379), rabbit anti-RFP (1:1000, Rockland 600-401-379), mouse anti-TH (1:1000, Millipore MAB5280), and rabbit anti-TH (1:1000; Abcam Ab112). The following secondary anti-bodies were used: donkey anti-chicken 488 (1:1000, Jackson ImmunoResearch 703-545-155), donkey anti-goat 488 (1:1000, ThermoFisher A-11055), donkey anti-rabbit 594 (1:1000, Jackson ImmunoResearch 711-587-003), donkey anti-rabbit 647 (1:1000, Jackson ImmunoResearch 711-605-152), donkey anti-mouse 594 (1:1000, Jackson ImmunoResearch 715-585-150), and donkey anti-mouse 647 (1:1000, Jackson ImmunoResearch 715-605-150). Brain sections were visualized, imaged, and processed using a Zeiss confocal microscope (Zeiss LSM 800).

### Quantification, statistical analysis, and reporducibility

All data processing and analyses were conducted using MATLAB and GraphPad Prism 8 (GraphPad Software Inc.). Data were tested for normality using the Kolmogorov-Smirnov test. When data met the normality assumption, paired or unpaired *t*-tests were used as appropriate. When data did not meet the normality assumption, Wilcoxon rank-sum or Mann-Whitney U tests were used as appropriate. To determine whether neuronal or dopamine sensor ROI data were significantly responsive toward behaviorally aligned events such as tones, the mean value of the continuous $\Delta F/F$ signal over a 1-s period after the behavioral timestamp was compared with the mean value 0.5 s before timestamp onset (Wilcoxon rank-sum test at $p < 0.05$). Neuronal ROI averaged tone responses were taken into consideration as a function of the behavioral training if it was deemed as responsive (i.e., having a statistically significant response; Wilcoxon rank-sum test at $p < 0.05$) during at least one of the sessions (habituation-conditioning or probe). The method for this characterization is based on prior analyses of published micro-endoscopic calcium imaging data and the observation of reproducible responses of auditory striatal neurons towards both passive and active engagement with tonal stimuli[7]. Similar statistical methods and thresholds have previously been used to analyze neuronal ROI's for responses towards a variety of behavior events such as tonal stimuli, movement, and escape behaviors[29,88–90]. Statistical methods were not utilized to predetermine appropriate sample sizes; rather, we used similar sample sizes as used in previous studies drawing similar biological conclusions[32,90].

Regarding the targeting of the auditory striatum, representative images of auditory striatum viral infusions, and reproducibility thereof, the written methods have optimized such experiments. Regarding optogenetic viral targeting of the auditory striatum, shown in Fig. 1 and the micrograph of Fig. 1B, 16 mice total are shown each with similar bilateral targeting and represent independent surgical and experimental biological replicates (4 experimental and 4 control for each respective intervention experiment). Regarding anterograde tracing shown in Fig. 2A, five independent replicates (out of 8 total attempts, three other attempts had either too much viral spread or insufficient infection) with similar cellular labeling in the auditory striatum and axonal labeling in the lateral amygdala was performed. Notably one replicate is shown in Fig. 2A and two others are shown in Supplemental Fig. 2B. Regarding retrograde tracing shown in Fig. 2B, there were 4 independent replicates that similarly labeled auditory striatal neurons when infusing the retrograde virus into the lateral amygdala (out of 6 attempts, with two others either having two much viral spread or insufficient infection). For dopamine imaging experiments as represented in Fig. 4E, 6 animals had similar viral/lens infusions (out of 7 attempts, one animal was excluded for having little or no live fluctuation in fluorescence). For optogenetic inhibition of nigrostriatal terminals as represented in Fig. 5B, 8 animals (out of 8

animals total, 4 per control and 4 per experimental) had similar fluorescent protein expression.

### Reporting summary

Further information on research design is available in the Nature Portfolio Reporting Summary linked to this article.

## Data availability

All data are provided in the main text figures or supplementary data in the Source Data file. Source data are provided with this manuscript. Source data are provided with this paper.

## Code availability

The custom codes used for data analysis in this study are available from the first author and/or corresponding author upon request.

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

## Acknowledgements

We thank Drs. Anissa Abi-Dargham, Robert Froemke, Adam Kepecs as well as Ge and Xiong laboratory members for their valuable comments on the manuscript. This work was supported by the National Institutes of Health (DC016746 and DC017470 to Q.X.; F30DCDC018214 to A.P.F.C.; NS089770, AG046875, and NS104868 to S.G.) and Stony Brook University internal fund (to Q.X.).

## Author contributions

A.P.F.C., S.G., and Q.X. designed the experiments. A.P.F.C. performed most experiments and data analysis. L.C., K.W.S., and E.C. performed behavioral imaging recordings and animal training. A.P.C., S.G., and Q.X. wrote the manuscript.

## Competing interests

The authors declare no competing interests.
