## [Peer Review File · Nature Communications]

Nigrostriatal dopamine modulates the striatal-amygdala pathway in auditory fear conditioningREVIEWER COMMENTS

Reviewer #1 (Remarks to the Author):

In this paper by Chen et al., the authors used an auditory fear conditioning task to assess the contribution of the tail of the striatum (TS) to fear memory. In the first experiment, they demonstrated that optogenetic inhibition of the TS during conditioning reduced mice's freezing response during the presentation of the auditory conditioned stimuli (CS) alone (probe session). Moreover, inhibition during the CS alone also reduced freezing, suggesting an involvement of the TS in fear conditioning learning and expression. The authors defined an unexplored pathway connecting the TS to the lateral amygdala (LA) both histologically and functionally. By genetically ablating the TS afference to the LA, the authors confirmed the involvement of this pathway in fear conditioning. They then used microendoscopic calcium activity recordings to investigate the mechanisms of how CS representations are encoded by TS neurons. Recordings showed a learning induced increase in activity to CS presentation during conditioning and following conditioning in the first probe session. Furthermore, the study showed that striatal dopamine may be a source of plasticity for this potentiation. By calcium imaging recording, the authors demonstrated that neurons of the substantia nigra pars compacta (SNc) projecting to the TS had increased CS responses during conditioning and the first probe session, and dopamine release in the TS followed a similar pattern. Finally, optogenetic inhibition of SNc dopaminergic afferents in the TS during conditioning reduced freezing during the two probe sessions. Finally, chemogenetic inhibition of SNc afferent in the TS impaired the development of TS neurons' responses to the CS.

This paper provides intriguing information about a new pathway connecting the posterior striatum to the lateral amygdala in the context of fear conditioning. However, there are important issues with the data presentation, missing control experiments and interpretation which make it difficult to definitively assess the manuscript. Details are listed below:

Major concerns:

1) For Fig. 2A and B, because of the close proximity of the auditory striatum and LA and the potential for viral spread between the two, the authors should assure the reader that viral spread between the two regions is not accounting for the labeling they see. This is particularly important as it appears that they used large injection volumes (650 nl for striatum, note that injection techniques and volumes for striatum and amygdala anatomical tracing were not described in the Methods section and this should be rectified). One thing they could do is to show tracings of the injection spread for anterograde and retrograde tracing experiments (for all injections or largest/smallest) along with histological images of sections to support the tracing). Something like this that strengthens the anatomical connectivity aspects of the paper is important, as this is a novel pathway they are describing.

2) For the caspase based lesioning study described in Fig. 2C and Suppl. Fig. 2C, there is a massive reduction (~80% it appears, Suppl. Fig 2C) in cell density in the striatum using retrograde CAV-cre in lateral amygdala and AAV-DIO-caspase in striatum. This is in contrast to smaller numbers of cells that are retrogradely labeled using a similar retrograde CAV viral approach (Fig. 2B). One possibility is that the AAV-DIO-caspase is not in fact cre-dependent and is killing striatal cells more generally. To assess this, the authors could determine whether this construct is expressed in the absence of the CAV-cre in lateral amygdala or, if there is not a fluorophore on this construct, determine cellular density in striatum using the same approach.

3) The authors demonstrated that auditory fear conditioning potentiates conditioned tone-induced activity in the TS using calcium imaging (Fig. 3). However, these results are presented without any behavioral information. The authors should report freezing behavior for animals in these experiments and explore whether changes in the response magnitude are correlated with different levels of animal freezing.

4) Also related to the imaging in Fig. 3, there are a number of potential issues and/or ways to improve the data presentation/analyses. First, Fig. 3C shows heat plots for CS responses across behavioral training session. However, it is not clear how these were aligned (i.e. from highest to lowest CS response? Realigned for each? Random alignment?). Ideally, they

should align individual cells to show highest to lowest CS response and realign for each session (i.e. Habituation, Conditioning, etc). In addition, they define cells as exhibiting increases or decreases in CS responses, but there is very little information given in the text or Methods about how this analysis was performed. It sounds like they used the $\Delta F/F$ response during the 1 sec period after CS onset and compared it to 0.5 sec before. More information is needed about how the mean values were calculated and used for statistics as well as a justification for looking at this time period. Additional analyses incorporating longer time periods (or different time periods) during CS presentation would also be helpful as it appears that there is some variation in the response across the CS period (e.g. excitation followed by inhibition, particularly during Conditioning and Probe 1; Excitation during the end of the CS in the Probe 1 trials possibly indicating prediction error responses). It would also be helpful to test whether these cells are shock responsive. Finally, to understand the population dynamics, a population averaged peri-event time histogram for all cells should be included for each session (Habituation, Conditioning, etc).

5) For the differential tone conditioning with imaging experiments presented in Suppl Fig. 3, the authors should also show behavioral freezing data to assess whether animals had selective freezing to the paired (5 kHz) vs. unpaired (10 kHz) tone. In addition, the number of animals used in these experiments should be listed. The authors should also think about adding this type of control group to later imaging experiments as well or discuss this as a shortcoming

6) Figure 4: The authors should reconcile the results from SNc recordings and the DA indicator. While the DA indicator showed a fast kinetic response after CS onset, SNr exhibited slower kinetics with prolonged activity over several seconds. Previous studies have shown that TS-projecting DA neurons coexpress the vesicular glutamate transporter 2 (VGLUT2), suggesting a corelease of DA and glutamate. Could this difference be explained by different neurotransmitter release?

7) Also related to Figure 4A-D, the approach the authors use to image from SNc neurons projecting to the striatum does not allow specific imaging of dopamine neurons and it is known that SNc also contains GABAergic projection neurons. The authors say that they

verified GCaMP6f was specific to dopamine neurons (and reference Suppl. Fig. 4B), but this was not in fact done. They could test this immunohistochemically by co-labeling GCaMP expressing cells for TH/DAT and a GABA marker, but my prediction would be that these neurons are not specifically dopamine. To make strong claims about imaging from dopamine projection neurons would likely require the use of a dual recombinase approach (e.g. retrograde virus from striatum expressing a cre-dependent flp recombinase in DAT-cre mice and a flp dependent GCaMP injecting into SNc). If they do see that the labeled SNc neurons are a mixed population but don't want to run the dual recombinase experiments then they should discuss this issue.

8) In Suppl. Fig. 4D, the lines denoting the bottom of the lens implantation are not visible. This line width, color and or zoom of the image should be adjusted to fix this.

9) Figure 5F and G: The authors showed that chemogenetic inhibition of SNc afferences to TS eliminates CS response development during conditioning in the TS. However, no animal behavior is presented. The authors should investigate whether CNO inhibition reduces freezing during conditioning and probe sessions.

10) Also related to Fig. 5A-D, a similar issue discussed in point 6 (above) arises here; there is no dopamine specificity to their viral approach. This could be dealt with using similar strategies to those described above.

11) The Materials and Methods section lacks information about chemogenetic inhibition, such as the time required for drug-designed receptor expression and the dose of CNO injected.

12) Fig 5 should show heat plots for the striatal neural activity for CNO-mCherry animals (in the main figure or supplement), as they do for the CNO-hM4Di group (Fig. 5F).

13) Calcium activity recordings of TS neurons provide a global picture of this structure's activity but do not allow us to understand the roles played by different populations of neurons/interneurons. Do MSNs expressing receptor D1 or D2 (direct or indirect pathway)

project differently onto the LA and exhibit different activity patterns? The anatomical questions here could be addressed using retrograde anatomical experiments and co-labeling for D1 and D2. Clarifying these points would provide a more complete understanding of the proposed pathway and its potential role in fear memory formation as well as providing a link to known cell types that have been extensively studied in other striatal regions.

14) Although the paper primarily focuses on auditory sensory integration, these experiments do not confirm that the highlighted processes are specific to auditory cue integration or more general stimulus-outcome association processes. The authors should control this parameter by using additional cue-fear conditioning experiences with other modalities (light cue, odor?) or moderate their claims about the auditory specificity of this pathway.

15) Related to the previous point, the authors focus on the auditory processing in this pathway, but almost all of their manipulations are performed during training and affect shock processing as well. In the one experiment they perform to inhibit CS evoked activity during the probe test (Fig. 1F-G) they see a minimal effect of the manipulation on freezing. An alternate possibility is that this pathway is important for conveying aversive shock information to the lateral amygdala. They could address this by looking more carefully at the shock responses in the striatal neurons and in the SNc inputs/dopamine responses (as suggested above), but my guess is that they will see shock responses throughout the system. Ideally, experiments using more temporally specific optogenetic inhibition during the shock period of training would be performed. Without this they should acknowledge this possibility in the Discussion.

16) The mechanism through which the striatal neurons regulate the lateral amygdala during learning are not clear and the authors should discuss this at more length. It seems they want to say that the striatal neurons projecting to amygdala transmit associative information about the auditory CS which boosts, in some way, lateral amygdala processing. If so, one question is how GABAergic (medium spiny) striatal neurons do this when the lateral amygdala neurons are known to develop excitation to the CS during fear conditioning (1). Thus it is not clear how this inhibitory striatal signal could facilitate lateral amygdala

excitatory associative processing. Of course, there are inhibitory networks in lateral amygdala which could be used for disinhibition and some recent studies report reductions in CS-evoked responding following conditioning, but the authors should address this point more clearly.

Minor:

1) Regarding freezing categorization, more information should be provided. Does the software differentiate between immobility (e.g., grooming) and freezing? Additionally, do all freezing quantifications cover the entire session duration (including CS+ intertrial periods), or are they only taken during CS presentation?

2) Fig1. B Legend: "Bilateral implantation and infusion of AAV-Cre with either AAV-DIO-GFP or AAV-DIO-ArchT". The schema suggests a virus injection of AAV-GFP and AAV-ArchT.

2) The calcium imaging activity graphs focus solely on neuron CS representation. Including information on neuron responses to the US during conditioning could provide additional valuable insights into how TS and DA neurons of the SNc encode sensory aversive information.

3) Fig4. B and F, Calcium imaging graph should have the same scale and same sizing to facilitate lecture and comparison.

4) Supplementary Fig.2: Histology picture of viral ablation of the auditory striatal of a representative animal should be appreciable.

5) Supplementary Fig.2: In the Fig.2 legend "Shade of red in intensities represent..." Must be corrected by blue.

Reference:

(1) Quirk, G. J., Reppas, J. B., LeDoux, J. E., & LaBar, R. S. (1995). Fear conditioning enhances short-latency auditory evoked potentials in humans. *Neuroreport*, 6(11), 1025-1029.

auditory responses of lateral amygdala neurons: parallel recordings in the freely behaving rat. *Neuron*, 15(5), 1029-1039.

Reviewer #2 (Remarks to the Author):

In this manuscript, Chen et al. demonstrate the role of the auditory striatum in classical auditory fear conditioning behavior in mice. The authors use a combination of circuit-specific opto- and chemogenetics as well as in vivo fluorescence imaging in freely behaving mice to show that the auditory striatum is necessary for proper fear memory formation. Mice show reduced freezing during future tone presentations when the auditory striatum was inhibited during conditioning and neuronal activity in the striatum is potentiated following fear conditioning. The authors further show that dopamine signaling in the auditory striatum is also significantly modulated following fear conditioning and might be responsible for striatal plasticity. This study finds that the role of the auditory striatum in fear memory is likely mediated by its projections to the lateral amygdala.

Overall, this manuscript highlights a novel and important behavioral role for the auditory striatum, a thus far understudied region of the brain. The experiments in this study are well-designed and the authors included many important control experiments. However, a couple of the findings warrant some additional experiments and/or analysis and discussion to substantiate the authors' claims.

- One curious aspect of the findings is that auditory striatum or dopaminergic input to the auditory striatum during conditioning does not affect freezing during the conditioning session, but only subsequent probe sessions. Potentiation of calcium signals in auditory striatum neurons, however, is observed during conditioning. How do the authors think about this? Does fear conditioning-induced striatal potentiation during the conditioning session also get abolished with chemogenetic inhibition of SNc inputs (Fig.5)?
- Looking at the degree of ablation in the caspase experiments (Supp. Fig. 2C) suggests that over 75% of neurons in the auditory striatum are projecting to the lateral amygdala (and thus receiving cre from CAV-cre). This seems rather high, and it might be necessary to add a control experiment showing the specificity of DIO-Caspase (AAV-DIO-Caspase injection without CAV-cre).

- Given the large percentage of neurons in the auditory striatum that is affected by viral Caspase ablation, it is difficult to conclude from this experiment that an auditory striatal projection to the lateral amygdala is the main circuit mediating the striatal role in auditory fear conditioning. Optogenetic inhibition/stimulation of projections from the auditory striatum in the amygdala would be more specific. Given the unusual direct projection from the striatum to the amygdala, this circuit warrants further characterization. For example, are D1 or D2 expressing neurons projecting to the amygdala? Are these just minor collaterals of more traditional striatal projection targets or is the amygdala indeed the main output of the auditory striatum? Please at least discuss.
- For statistical quantification of freezing in Fig. 1 E and 1G the authors seem to use the response to individual tones as individual data points, however, the authors should average the response for each mouse and treat individual mice as data points for analysis since multiple responses to tones in the same mouse are not independent observations. A similar issue seems to exist for calcium responses in Fig. 4D and H.
- In general, it is not clear what individual data points plotted in many figures represent. For example, in Fig. 4C and D it is suggested that each circle is an individual neuronal ROI, and in Fig 4 G and H in each point should be an individual GRIN FOV. However, the number of circles does not match the n number stated in the legends.
- The authors include a great control experiment testing for calcium responses to a tone not paired with shocks (Supp. Fig. 3A-C), however, it would be important to also show the calcium response to the paired tone (Tone A) for this set of mice.
- Authors should show the freezing behavior for mice in which calcium imaging in the auditory striatum was performed (Fig. 3 and 5E-G), similar to what was done for SNc recordings.
- What are the criteria used to determine which of the detected ROIs to include in the calcium image analysis? E.g., Fig.4D says that only 23 out of 58 detected neuronal ROIs are plotted.

Minor points

- Text in results mentions that freezing was $12.32 \pm 1.02\%$ during habituation, however in Fig. 1C it looks like the average should be below 10%.
- The results state that all tones are interspersed with 100- to 180-s intertrial intervals, but the methods state 60-100 s intertrial intervals.
- Please state the diameter of optic fiber cannula and GRIN lens implants used.
- In some figures the numbers of mice/neurons are missing (e.g., Supp. Fig. 1G, Supp. Fig. 3).

Reviewer #3 (Remarks to the Author):

In this study Chen and colleagues explored the function of the auditory striatum (or the “tail of the striatum”) in auditory fear conditioning. They report that optogenetically inhibiting auditory striatal neurons impaired fear memory formation, which was mediated through the striatal-lateral amygdala pathway. Furthermore, they used calcium imaging in behaving mice to show that auditory striatal neuronal responses to conditioned tones were potentiated across memory acquisition and expression. They also provided evidence suggesting that nigrostriatal dopaminergic projections may play a role in modulating conditioning-induced striatal potentiation.

This study used a combination of cutting-edge technologies to convincingly show the plasticity of auditory striatum neuron response plasticity and its involvement in fear conditioning. In addition, the study provides nice evidence suggesting the involvement of a nigro-striatal-amygdala circuit in auditory fear conditioning. These findings are novel and further our understanding of the neural circuits underlying fear conditioning. In general, I think this paper fits nicely to Nature Communications. I have some suggestions that may strengthen the major conclusions.

1. The authors propose a nigro-striatal-amygdala circuit for auditory-conditioned fear memory formation and expression. It would be nice to assess the flow of information in this circuit in their imaging data. For example, do dopamine neurons fire earlier than neurons in auditory striatum in response to CS during conditioning or memory retrieval? From the traces in Figure 4B and those in Figure 3E, it seems that there is a longer latency for

dopamine neurons. However, this could be skewed by averaging data, so a cell-by-cell analysis on response latency is needed for each of these two populations. It would be interesting to see if the distributions of response latencies are different or not.

2. Related to #1 above, it seems that the dopamine neuron responses (Figure 4B) had a longer latency than the dopamine sensor responses (Figure 4F). Again, a cell-by-cell analysis on dopamine neuron response latency is needed, to see whether some dopamine cells had short latencies.

3. The authors need to provide evidence that the retrograde strategy shown in Figure 4A and Figure 5E actually targeted dopamine neurons in the SNc.

4. Supplementary Figure 2: please provide representative histology images to show the viral ablation effect.

5. Page 15, last paragraph: avoid using “imaging” to describe photometry data, as photometry is not imaging.

Response to reviewers' comments

We thank the reviewers for their encouraging and constructive comments. To address these concerns, we have performed additional experiments and new analyses. We have extensively revised the text accordingly including the addition of discussions as suggested by reviewers. The following are point to point responses (reviewers' comments are italic).

Reviewer #1:

In this paper by Chen et al., the authors used an auditory fear conditioning task to assess the contribution of the tail of the striatum (TS) to fear memory. In the first experiment, they demonstrated that optogenetic inhibition of the TS during conditioning reduced mice's freezing response during the presentation of the auditory conditioned stimuli (CS) alone (probe session). Moreover, inhibition during the CS alone also reduced freezing, suggesting an involvement of the TS in fear conditioning learning and expression. The authors defined an unexplored pathway connecting the TS to the lateral amygdala (LA) both histologically and functionally. By genetically ablating the TS afference to the LA, the authors confirmed the involvement of this pathway in fear conditioning. They then used microendoscopic calcium activity recordings to investigate the mechanisms of how CS representations are encoded by TS neurons. Recordings showed a learning induced increase in activity to CS presentation during conditioning and following conditioning in the first probe session. Furthermore, the study showed that striatal dopamine may be a source of plasticity for this potentiation. By calcium imaging recording, the authors demonstrated that neurons of the substantia nigra pars compacta (SNc) projecting to the TS had increased CS responses during conditioning and the first probe session, and dopamine release in the TS followed a similar pattern. Finally, optogenetic inhibition of SNc dopaminergic afferents in the TS during conditioning reduced freezing during the two probe sessions. Finally, chemogenetic inhibition of SNc afferent in the TS impaired the development of TS neurons' responses to the CS. This paper provides intriguing information about a new pathway connecting the posterior striatum to the lateral amygdala in the context of fear conditioning. However, there are important issues with the data presentation, missing control experiments and interpretation which make it difficult to definitively assess the manuscript. Details are listed below:

Major concerns:

1) For Fig. 2A and B, because of the close proximity of the auditory striatum and LA and the potential for viral spread between the two, the authors should assure the reader that viral spread between the two regions is not accounting for the labeling they see. This is particularly important as it appears that they used large injection volumes (650 nl for striatum, note that injection techniques and volumes for striatum and amygdala anatomical tracing were not described in the Methods section and this should be rectified). One thing they could do is to show tracings of the injection spread for anterograde and retrograde tracing experiments (for all injections or largest/smallest) along with histological images of sections to support the tracing). Something like this that strengthens the anatomical connectivity aspects of the paper is important, as this is a novel pathway they are describing.

We appreciate this line of suggestions and agree with the reviewer that extensive characterization of this pathway would be necessary and strengthen the manuscript. In the revision, as suggested, we included histological images representing the largest and smallest viral spread in the injection sites (**Suppl. Fig. 2A**) and the corresponding tracing results to ensure 'spreading' was not the cause. We revised the **Material and Methods** section to include injection details (page 24). These are indeed important points, and we revisited the whole text and provided experimental details as briefed below.

Clarification for when small volume injections were used: in experiments using genetically non-restricted (straightforward) viruses, we injected smaller volumes to anatomically constrain our targeting of both the amygdala and the auditory striatum (250-300 nl for both the auditory striatum and lateral amygdala). This is relevant for injections performed in Figure 1-2.

Clarification for when large volume injections were used: For microendoscopic imaging (cre-dependent and straightforward) and optogenetic experiments (cre-dependent), we as well as other labs, have had

greater imaging/optogenetic success by injecting larger volumes (500-650 nl). Thus, depending on the specific experiment in the manuscript, we adjusted our viral injection volumes and as with all experiments verified specific targeting on post-hoc inspection. In our updated manuscript, we include all the specific viral injection volumes and details in the methods. This is relevant for Figure 3-5.

Together with the above clarifications, we included representative images demonstrating the spread and variation of viral expression. In all experiments, we verified on post-hoc inspection whether there was notable contamination or ablation of neighboring regions outside of the striatum. In the **Figure 2** we showed examples with medium spread of virus in the injection site, and below (also included in **Suppl. Fig. 2A**) we included two additional examples showing the largest and smallest spread of virus in injection sites for our anterograde eGFP tracing experiments (**R1Q1**). For both examples, we observed GFP+ cell bodies in the auditory striatum but not amygdala, whereas there are observable GFP+ axonal terminals in the lateral amygdala.

R1Q1 (Suppl. Fig. 2A) Anterograde tracing from the auditory striatum to the lateral amygdala. Two sets of anterograde tracing images exhibit the largest and smallest source spread when injecting a target volume of 250-300 nl tracing virus (AAV8-eGFP) into the auditory striatum. **Left**, example of largest spread. Left is a 10X image of injection site; middle is zoom in image of injection site demonstrating auditory striatum and lateral amygdala border; right is a 40X image exhibiting cell bodies in the auditory striatum and axons in the lateral amygdala. **Right**, example of smallest spread. Left is a 10X image of injection site; middle is zoom in image of injection site demonstrating auditory striatum and lateral amygdala border; right is a 40X image exhibiting cell bodies in the auditory striatum and axons in the lateral amygdala. Scale bars are 400 μ m for 10X injection site images and 100 μ m for 40X images.

2) For the caspase based lesioning study described in Fig. 2C and Suppl. Fig. 2C, there is a massive reduction (~80% it appears, Suppl. Fig 2C) in cell density in the striatum using retrograde CAV-cre in lateral amygdala and AAV-DIO-caspase in striatum. This is in contrast to smaller numbers of cells that are retrogradely labeled using a similar retrograde CAV viral approach (Fig. 2B). One possibility is that the AAV-DIO-caspase is not in fact cre-dependent and is killing striatal cells more generally. To assess this, the authors could determine whether this construct is expressed in the absence of the CAV-cre in lateral amygdala or, if there is not a fluorophore on this construct, determine cellular density in striatum using the same approach.

We thank the reviewer for pointing out this confusion, which was from the mistake that we mislabeled the unrestricted ablation results. In the revision, we have corrected the mistake by providing representative images, quantifications, and corresponding behavioral results for both unrestricted ablation and projection-specific ablation in the auditory striatum. We included these results in **Suppl. Fig. 2** and have modified the text accordingly on pages 8-9.

In brief, unrestricted ablation of neurons in the auditory striatum was performed by injecting AAV-Caspase into the auditory striatum (**Suppl. Fig. 2B**). As mentioned by the reviewer, projection-specific ablation of auditory striatal neurons was performed by injecting CAV-Cre into lateral amygdala and

AAV-DIO-Caspase into the auditory striatum (**Fig. 2C**). Both ablation strategies led to reduced freezing behaviors in the auditory fear conditioning (**Fig. 2C & Suppl. Fig. 2C**).

In unrestricted ablation, there is a large reduction of neurons within the auditory striatum, while sparing those in the lateral amygdala and neighboring regions (**Suppl. Fig. 2D&E**). The projection-specific ablation caused less reduction of auditory striatal neurons (**Suppl. Fig. 2F&G**, 32% reduction).

R1Q2 (Suppl. Fig. 2B-G) Caspase-mediated ablation of the auditory striatal neurons. B. Schematic for ablation of the auditory striatum without genetic restriction. **C.** Freezing behavior across cohorts of control vs. ablation. $n = 5$ mice per group, error bars are SEM (unpaired Mann-Whitney test; ns $p > 0.05$, * $p < 0.05$; ** $p < 0.01$). **D&E**, unrestricted ablation. **F&G**, projection-specific ablation. **D.** Images, auditory striatum in control animal with lateral amygdala shown, and comparable image with caspase ablation. The bottom two panels are respective zoomed-in images. Right graph, Quantification of neuronal density in the auditory striatum between control and caspase groups for unrestricted ablation Control, 1.10 ± 0.02 ; Ablation, 0.20 ± 0.03 neuron $\times 10^3/\text{mm}^3$. Error bars are SEM (unpaired Mann-Whitney test; **** $p < 0.0001$). **E.** Schematic of viral spread and ablation spread for both control and ablation groups for experimental data shown in **D**. **F.** Histology of amygdala-projecting auditory striatum neuronal ablation experiment performed in **Fig. 2**. Images, auditory striatum in control animal with lateral amygdala shown, and comparable image with projection-specific caspase ablation. The bottom two panels are respective zoomed-in images. Graph, quantification of neuronal density in the auditory striatum between control and caspase groups for projection-specific ablation. Control, 1.07 ± 0.04 ; Ablation, 0.72 ± 0.08 neuron $\times 10^3/\text{mm}^3$. Error bars are SEM (unpaired Mann-Whitney test; **** $p < 0.0001$). **G.** Schematic of viral spread and ablation spread for both control and ablation groups for experimental data shown in **F**. For histological images, labeled in blue is DAPI and green is the neuronal marker NeuN. Scale bars are $200 \mu\text{m}$ and $100 \mu\text{m}$ for overall and zoomed-in images, respectively.

3) The authors demonstrated that auditory fear conditioning potentiates conditioned tone-induced activity in the TS using calcium imaging (Fig. 3). However, these results are presented without any behavioral information. The authors should report freezing behavior for animals in these experiments and explore whether changes in the response magnitude are correlated with different levels of animal freezing.

We agree with the reviewer that these analyses are important. In the revised manuscript, we included the data in **Suppl. Fig. 3F-I** presenting the quantification of freezing behaviors and the correlation between freezing and tonal response. These results are added into the Result section (pages 11-12). As suggested by the reviewer, although there were overall increased freezing behaviors after conditioning, mice exhibited a distribution of freezing when we calculated individual freezing time towards each tone presentation (**Suppl. Fig. 3F-H**). To explore whether neuronal tonal responses are correlated with freezing levels, we calculated individual neuronal responses to tones (averaged peak Zscore $\Delta F/F$ from all responsive probe sessions) and plotted them against the corresponding freezing percentages in the same sessions. Using Pearson correlation, we found that there is a moderate positive correlation between the tonal responses and freezing levels (**Suppl. Fig. 4I**, $R = 0.3105$; $p = 1.88 \times 10^{-5}$). Thus, this analysis indeed suggests that auditory striatal tonal responses correlate with learned freezing behaviors.

R1Q3 (Suppl. Fig. 3F-I). Freezing behavior of animals during microendoscopic neuronal recordings. **F**, Freezing in response to tones during habituation, conditioning, and probe sessions. Freezing percentages for two consecutive tones were averaged as a single data point. Semi-transparent data points present each individual animal; each individual shape represents a different animal. Overall averaged data are stylized as solid. Error bars are standard error of the mean (SEM; $n = 6$ mice). **G**, Semi-transparent data points present each individual animal; each individual shape represents a different animal. Overall averaged data are stylized as solid. Error bars are standard error of the mean (SEM; $n = 6$ mice). **H**, Distributions of freezing behaviors (left) and neuronal tonal responses (right) to individual tones in probe sessions. **I**, Correlation between the freezing behaviors and neuronal tonal responses. Pearson correlation, $R = 0.3105$; $p = 1.88 \times 10^{-5}$. $n=188$ neurons from 6 mice.

4) Also related to the imaging in Fig. 3, there are a number of potential issues and/or ways to improve the data presentation/analyses. First, Fig. 3C shows heat plots for CS responses across behavioral training session. However, it is not clear how these were aligned (i.e. from highest to lowest CS response? Realigned for each? Random alignment?). Ideally, they should align individual cells to show highest to lowest CS response and realign for each session (i.e. Habituation, Conditioning, etc). In addition, they define cells as exhibiting increases or decreases in CS responses, but there is very little information given in the text or Methods about how this analysis was performed. It sounds like they used the $\Delta F/F$ response during the 1 sec period after CS onset and compared it to 0.5 sec before. More information is needed about how the mean values were calculated and used for statistics as well as a justification for looking at this time period. Additional analyses incorporating longer time periods (or different time periods) during CS presentation would also be helpful as it appears that there is some variation in the response across the CS period (e.g. excitation followed by inhibition, particularly during Conditioning and Probe 1; Excitation during the end of the CS in the Probe 1 trials possibly indicating prediction error responses). It would also be helpful to test whether these cells are shock responsive. Finally, to understand the population dynamics, a population averaged peri-event time histogram for all cells should be included for each session (Habituation, Conditioning, etc).

We appreciate the reviewer's suggestions. In the revised manuscript, we 1) sorted individual cells' activities from highest to lowest CS responses and realigned it for each session; 2) added the PSTH plots for the population average for each session. 3) provided analysis details on changes of CS responses across sessions, and the justification of the chosen time window; 4) added additional analysis using longer time window; 5) added a new plot demonstrating cells' responses to foot shock.

We acknowledge the cell sorting suggestion. In the previous version, cells were sorted based on the order of experiments. In the revised **Fig. 3C**, we've sorted the cells from highest to lowest CS responses based on the calculations of peak calcium signals within a 1 s time-window for each session. Averaged population activity for each session is also plotted below the heatmap. These new plots demonstrate more clearly the changes of CS responses in conditioning and probe sessions.

Regarding the time windows we used to calculate CS response, we followed the strategy we used in one previous study (Chen, et al 2022) which was based on the observation of reproducible auditory striatal responses to tones in both passive and engaged contexts. These criteria and statistical analysis are similar to other labs analyzing responses towards tonal stimuli and a variety of behaviors such as movement and escape (Li et al., 2021; Parker et al., 2018; Howe et al., 2016). We have included a description of the justification for this statistical definition and criteria in the methods of the manuscript (page 33). Also copied here: "To determine whether neuronal or dopamine sensor ROI data were significantly responsive toward behaviorally aligned events such as tones, the mean value of the continuous $\Delta F/F$ signal over a 1 s period after the behavioral timestamp was compared with the mean value 0.5 s before timestamp onset (Wilcoxon rank-sum test at $p < 0.05$). Neuronal ROI averaged tone responses were taken into consideration as a function of the behavioral training if it was deemed as responsive (i.e., having a statistically significant response; Wilcoxon rank-sum test at $p < 0.05$) during at least one of the sessions (habituation-conditioning or probe). The method for this characterization is based on prior analyses of published microendoscopic calcium imaging data and the observation of reproducible responses of auditory striatal neurons towards both passive and active engagement with tonal stimuli [7]. Similar statistical methods and thresholds have previously been used to analyze neuronal ROI's for responses towards a variety of behavior events such as tonal stimuli, movement, and escape behaviors [29, 71-73]".

However, given that the CS used here is 20 seconds long, and from the heatmaps and PSTH the response peaks are mostly within a 5 s time window. We performed the same analysis using a 5 s time window. In below **R1Q4B**, we showed that although the absolute numbers varied depending on the time windows (1 s or 5 s) we used to calculate the responses percentages, the overall changes across different sessions are the same. In the revised manuscript, we used the analysis of 1 s time window in **Fig. 3** and added the 5 s analyses in **Suppl. Fig. 4A**.

We added additional analysis on whether auditory striatal neurons respond to foot shock stimuli (**Suppl. Fig. 4B**). Using similar criteria as above, a large portion of neurons (32.38%) responded to foot shock stimuli (left). However, if we sorted these neurons based on their CS responses and extended the time course to include their foot shock responses, we found not many of CS-responsive neurons are shock responsive (right).

Figure 3

Supplementary Figure 4

R1Q4 (Fig. 3C and Suppl. Fig. 4) The auditory striatal neuronal responses to conditioned tones, and neuronal tonal responses are correlated with freezing levels. **Fig. 3C**, Upper row, heatmaps of neuronal ROI responses ($n = 262$ neurons from 6 mice) to tones during habituation, conditioning, and probe sessions. Middle row, averaged neuronal ROI responses for the corresponding sessions. Bottom row, quantification of the proportion of neurons with increased (black), decreased (dark gray), or no significant change (light gray; Wilcoxon rank-sum test, $p > 0.05$) in response to tones. **Suppl. Fig. 4A**, Using 5 s as the time window, the quantification of the proportion of neurons with increased (black), decreased (dark gray), or no significant change (light gray; Wilcoxon rank-sum test, $p > 0.05$) in response to tones. **B**, Left, heatmap of auditory striatal neuronal responses towards foot shock. Right, the same neuronal ROIs arranged by tonal response amplitude with extended time course to include responses to foot shocks starting at the 20 s time point. $n = 262$ neuronal ROIs from 6 mice.

5) For the differential tone conditioning with imaging experiments presented in Suppl Fig. 3, the authors should also show behavioral freezing data to assess whether animals had selective freezing to the paired (5 kHz) vs. unpaired (10 kHz) tone. In addition, the number of animals used in these experiments should be listed. The authors should also think about adding this type of control group to later imaging experiments as well or discuss this as a shortcoming.

We agreed with the reviewer and added the behavioral data and number of mice in **Suppl. Fig. 3A-D**. We also added the same type of control for DA sensor recordings in **Suppl. Fig. 5E&F** and discussed this shortcoming for SNc imaging experiment (page 18).

In the revised figures (also shown below), we included the freezing behavioral quantifications for both conditioned tone (Tone A) and unconditioned tone (Tone B), from the mice used for striatal calcium imaging (**Suppl. Fig. 3D**) and from mice used for DA sensor recording (**Suppl. Fig. 5E**). From both groups we found that the mice selectively freeze to the conditioned tones. Consistent with a lack of behavioral response towards Tone B, we found that DA responses towards unconditioned tones do not potentiate either (**Suppl. Fig. 5F**). These control experiments suggest that the striatal and dopaminergic potentiation observed in **Figs. 3&4** are likely due to tone-shock pairings. However, these control experiments were not performed for all imaging (i.e. SNc calcium imaging) and optogenetic datasets due to experimental constraints, therefore we add to our discussion this possible limitation:

“The potentiation towards tones appears to be conditioning-specific, as we did not observe potentiation towards unpaired tones in control experiments (**Supple. Figs. 3C&5F**). However, we note here that for SNc neuronal imaging, we did not perform the Tone B control experiment and therefore there could be individual neurons that potentiate based on context or generalized threat processing.”

Supplementary Figure 3

Supplementary Figure 5

R1Q5 Responses to unconditioned tones (Suppl. Figs. 3D & 5E&F). **D**, Corresponding behavioral data for striatal neuronal imaging analysis. Left, averaged freezing percentage in response to tone A during habituation, conditioning, and probe sessions. Right, averaged freezing in response to tone B during pre-conditioning and post-conditioning sessions (3 post-conditioning tone B sessions; $n = 4$ mice). **E**, Same as D but for striatal DA sensor mice. ($n = 4$ mice). **F**, Averaged and individual auditory striatal DA responses towards Tone A (left) or Tone B (right) during habituation, conditioning, and probe sessions. Individual dots represent averaged responses per two consecutive tones across three mice. Error bars are SEM (Wilcoxon rank-sum test; ns, $p > 0.05$; *** $p < 0.001$; $n = 3$ mice).

6) Figure 4: The authors should reconcile the results from SNc recordings and the DA indicator. While the DA indicator showed a fast kinetic response after CS onset, SNr exhibited slower kinetics with prolonged activity over several seconds. Previous studies have shown that TS-projecting DA neurons coexpress the vesicular glutamate transporter 2 (VGLUT2), suggesting a corelease of DA and glutamate. Could this difference be explained by different neurotransmitter release?

We thank the reviewer for pointing out this confusion. In the revision we updated representative traces and populational analyses to address this concern (Fig. 4B&C and Suppl. Fig. 5G) and added a discussion to acknowledge the technical limitation and alternative interpretations (pages 20-21).

We replaced the Fig. 4B example traces with shorter latency that better represent the majority of SNc neuronal responses. We also included a heatmap to demonstrate all tone responsive SNc neuronal responses (Fig. 4C or see below R1Q6). Indeed, there are a few SNc neurons that displayed substantially long latency. In fact, most SNc neuronal responses have no statistical difference compared to the DA activity in the auditory striatum but significantly shorter than the striatal neuronal responses (Suppl. Fig. 5G or see below R1Q6).

The kinetics and peak of fluorescence between GCaMP6f (SNc neuronal activity) and DA2m (DA activity) are different and the onset of rise in fluorescence may not be a sufficient quantitative indicator of activity onset (i.e., real time calcium fluctuation may be sufficient to trigger DA release and such a change may not be reflected by *in vivo* imaging). Thus, from our results we do not know whether striatal DA activity is faster than the SNc somatic activity or not. As the reviewer mentioned, it is possible that VGLUT2 activity may locally trigger dopaminergic release from axon terminals. Another possibility could be local cholinergic activation. The reviewer raised a very exciting possibility of being tested for the next phase of this exploration. We added a discussion to include the limitations and possibilities.

Figure 4

R1Q6 (Fig. 4C & Suppl. Fig. 5G). C, Averaged tonal response latencies for individual SNc neuronal ROI (n = 23 neuronal ROI from 6 mice). **G**, Comparison of tonal response latencies from SNc neuronal ROI, DA sensor FOV (n = 6 mice), and auditory striatal neuronal ROI (n = 188 from 6 mice). Error bars are SEM (Mann-Whitney test; ns, p > 0.05; **p < 0.01).

Supplementary Figure 5

7) Also related to Figure 4A-D, the approach the authors use to image from SNc neurons projecting to the striatum does not allow specific imaging of dopamine neurons and it is known that SNc also contains GABAergic projection neurons. The authors say that they verified GCaMP6f was specific to dopamine neurons (and reference Suppl. Fig. 4B), but this was not in fact done. They could test this immunohistochemically by co-labeling GCaMP expressing cells for TH/DAT and a GABA marker, but my prediction would be that these neurons are not specifically dopamine. To make strong claims about imaging from dopamine projection neurons would likely require the use of a dual recombinase approach (e.g. retrograde virus from striatum expressing a cre-dependent flp recombinase in DAT-cre mice and a flp dependent GCaMP injecting into SNc). If they do see that the labeled SNc neurons are a mixed population but don't want to run the dual recombinase experiments then they should discuss this issue. We thank the reviewer for pointing out this issue. Indeed, the striatal-projecting SNc neurons we imaged may not be purely dopaminergic. As suggested by the reviewer, we performed immunohistochemistry staining to confirm these neurons' identity, revised our statement, and added discussion.

We injected the retrograde labeling virus CAV expressing mCherry (CAV-mCherry) into the auditory striatum and identified the mCherry+ neurons in the SNc (**R1Q7A**). We then performed TH staining to identify dopaminergic neurons in the SNc and quantified the numbers of TH+ neurons and mCherry+ neurons. We found that $94.8 \pm 1.92\%$ mCherry+ neurons are TH+ and $5.4 \pm 2.07\%$ neurons are TH- (**R1Q7B**), consistent with our previous published work (**Fig. 1**, (Chen et al., 2022)). We made a statement that the majority of SNc neurons projecting to the auditory striatum are dopaminergic. We included this new analysis in **Suppl. Fig. 5A&B**. Accordingly, we revised the statement on pages 12-13. However, we do agree that the labeling strategy in this recording would not preclude the small portion of non-dopaminergic neurons (~5% according to the above tracing quantification). In the resubmission, we have added a short discussion on this limitation (page 20). We will be thrilled to try the dual recombinase experiments, which my laboratory at this moment does not have the tools, for our next phase experiments on this project.

R1Q7 (Suppl. Fig. 5A&B). Retrograde tracing from the auditory striatum primarily labels dopaminergic neurons in the substantia nigra pars compacta (SNc). **A.** Left upper, injection schematic of CAV-mCherry into the auditory striatum. Left bottom, representative imaging of injection site and traversing mCherry+ axons in the auditory striatum. Right, retrograde labeling in midbrain dopamine regions, the VTA and SNc. mCherry is pseudo-colored in gold, with TH labeling in blue. Scale bar is 100 μ m. **B.** The left two bars show percentage quantification of mCherry+ neurons in the VTA and SNc. The right two bars show the percentage of mCherry+ neurons that are either TH- or TH+ neurons.

8) In Suppl. Fig. 4D, the lines denoting the bottom of the lens implantation are not visible. This line width, color and or zoom of the image should be adjusted to fix this.

We have revised the figure (now as **Suppl. Fig. 5C&D**) to fix the visibility issue.

9) Figure 5F and G: The authors showed that chemogenetic inhibition of SNc afferences to TS eliminates CS response development during conditioning in the TS. However, no animal behavior is presented. The authors should investigate whether CNO inhibition reduces freezing during conditioning and probe sessions.

We thank the reviewer for pointing out this missing information. Indeed, we have tested whether CNO inhibition affects freezing behaviors after conditioning. In the revised **Fig. 5G**, we added the behavioral quantifications to show that chemogenetically inhibiting SNc neurons projecting to the auditory striatum impaired fear conditioning.

R1Q9 (Fig. 5G) Line plots of averaged freezing percentage in response to tones during habituation, conditioning, and probe sessions. Error bars are SEM, n = 5 mice from each group (unpaired Mann-Whitney test; ns, $p > 0.05$; ** $p < 0.01$; *** $p < 0.05$).

10) Also related to Fig. 5A-D, a similar issue discussed in point 6 (above) arises here; there is no dopamine specificity to their viral approach. This could be dealt with using similar strategies to those described above.

We thank the reviewer again for pointing out this issue. Similar as above (#7), we performed immunohistochemistry and quantification to confirm that the majority of striatal-projecting SNc neurons are dopaminergic (**R1Q7**), revised the statement (pages 12-13) and included a discussion on this limit (page 20).

11) The Materials and Methods section lacks information about chemogenetic inhibition, such as the time required for drug-designed receptor expression and the dose of CNO injected.

In the revised manuscript, we have included the protocol for chemogenetic inhibition in **Materials and Methods** section (pages 29-30). Also copied below:

“Chemogenetic Inhibition Simultaneous microendoscopic imaging and chemogenetic inhibition were performed similar to behavioral experiments performed in a prior publication (Chen et al., 2022). Briefly, in wild-type mice, viral infusion of CAV-Cre to the bilateral auditory striatum, AAV-GCaMP6f to the left auditory striatum, and cre-dependent AAV-hM4Di to the SNc, was performed in one surgery. Simultaneously, lens implantation to the left auditory striatum was performed. Mice were allowed to recover for at least 3 weeks before baseplate implantation. After recovery, mice were then habituated to the experimenter through behavioral handling for one day. Mice were mounted with the endoscopic camera and allowed to habituate to the camera during free movement. Subsequently, the animals were habituated to i.p. injections of 300 μ l water (vehicle) for three days. After habituation towards handling and i.p. injections, one mouse cohort dedicated to mice vehicle and one mouse cohort dedicated to CNO (Enzo; i.p. 2.5 mg/kg, 300 μ l diluted in water) were subjected to injections 30 minutes prior to fear conditioning and simultaneous in vivo imaging sessions.”

12) Fig 5 should show heat plots for the striatal neural activity for CNO-mCherry animals (in the main figure or supplement), as they do for the CNO-hM4Di group (Fig. 5F).

As suggested, we included the plots in the revised Figure 5. Also shown below:

R1Q12 (Fig. 5E). Heatmap of striatal neuronal activity aligned to tone onset during habituation and conditioning sessions for mice undergoing CNO-mediated nigrostriatal chemogenetic inhibition (n = 72 neuronal ROIs across 5 hM4Di mice, and n = 59 neuronal ROIs across 5 mCherry mice).

13) Calcium activity recordings of TS neurons provide a global picture of this structure's activity but do not allow us to understand the roles played by different populations of neurons/interneurons. Do MSNs expressing receptor D1 or D2 (direct or indirect pathway) project differently onto the LA and exhibit different activity patterns? The anatomical questions here could be addressed using retrograde anatomical experiments and co-labeling for D1 and D2. Clarifying these points would provide a more complete understanding of the proposed pathway and its potential role in fear memory formation as well as providing a link to known cell types that have been extensively studied in other striatal regions. We agree with the reviewer that the original results do not differentiate the activities of the two major populations (D1 and D2 MSNs) in the auditory striatum. Following the suggestion, we performed anterograde viral tracing experiments and found that both D1 and D2 MSNs project to the lateral amygdala. In brief, we injected AAV-DIO-tdTomato into the auditory striatum of D1-Cre and D2-Cre (A2a-Cre) mice. Infected neurons were found in the auditory striatum identified as D1-MSNs or D1-MSNs, respectively. The tdTomato+ axon terminals (pseudo-colored in green for a better visualization) were found in the lateral amygdala of both cohorts of mice. This seems to be very interesting. Of course, further studies will need to be done to examine whether they exhibit differential activity patterns in tone-cued fear conditioning. We added this limitation in the **Discussion** section (page 17, also copied below).

“...Furthermore, our results do not differentiate the activities of striatal neuronal populations (e.g. D1 and D2 MSNs). Anterograde viral tracing suggests that both D1 and D2 MSNs project to the lateral amygdala. How dopaminergic modulation through these neurons may impact auditory fear conditioning? In the literature, both D1 and D2 receptors systemic pharmacological antagonism has been shown to abrogate different forms of fear conditioning with mixed interpretations (De Bundel et al., 2016; de Souza Caetano et al., 2013; Greba & Kokkinidis, 2000; Guarraci et al., 1999; Ikegami et al., 2014; Inoue et al., 2000; Ponnusamy et al., 2005). Differential roles for the D1 and D2 MSN pathways within the dorsal and ventral striatal subdivisions have been suggested that the pathways may correspond to appetitive-positive-actions or aversive-negative-inhibitions depending on the behavior being studied (Gerfen, 2022; Isett et al., 2023; Nishioka et al., 2023). However, this D1 and D2 opponency is nuanced, with evidence suggesting that the D1 pathway supports generalized learning while the D2 pathway could aid in refinement of learning (Iino et al., 2020; Matamales et al., 2020). In the auditory striatum, both D1 and D2 MSNs are responsive towards auditory cues (Nardoci et al., 2022). Our previous works indicate that the D1 pathway is important for auditory discrimination and task performance, while the D2 pathway appears to be dispensable (Chen et al., 2022). Future studies are needed to examine the potential functional differences of D1 and D2 MSNs in fear conditioning.”

R1Q13. Both D1 and D2 MSNs in the auditory striatum project to the lateral amygdala. **A**, histology of anterograde tracing from the auditory striatum in a D1-Cre mouse with Cre-dependent AAV-tdTomato (pseudo-color green). Left, representative image of the injection site. Right, zoomed-in images of infected cell bodies in the auditory striatum and axon terminals in the lateral amygdala. **B**, same as in A but from a D2-Cre (A2a-Cre) mouse. Scale bars for large-view images and zoomed-in images are 200 μm and 20 μm , respectively.

14) Although the paper primarily focuses on auditory sensory integration, these experiments do not confirm that the highlighted processes are specific to auditory cue integration or more general stimulus-outcome association processes. The authors should control this parameter by using additional cue-fear conditioning experiences with other modalities (light cue, odor?) or moderate their claims about the auditory specificity of this pathway.

We agree with the reviewer that the current dataset cannot be concluded specific to auditory stimulus. As suggested and to incorporate this important point, we revised our statements and also added a short discussion for clarification.

We added the discussion (page 19) as the following: “We note here that while this study utilized auditory cues as with prior studies (A. P. F. Chen et al., 2022; L. Chen et al., 2019; Li et al., 2021; Xiong et al., 2015; Znamenskiy & Zador, 2013), this tail portion of the striatum receives projections from multiple sensory channels including auditory, visual, and somatosensory cortical inputs (Hintiryan et al., 2016; Hunnicutt et al., 2016; Valjent & Gangarossa, 2021). Thus, while we only tested auditory stimulus, this region may facilitate the learning of multiple sensory modalities as has been shown for the amygdala itself (Bergstrom & Johnson, 2014; Hakim et al., 2019).”

We also revised the sentence (page 21) “The auditory striatum similarly receives convergent input from these sensory regions, and its function appears to depend on medial geniculate body and auditory cortical inputs” – to instead of “depend on” to “be modulated by.” We made this change to reflect the notion that the auditory striatum may be modulated by a variety of sensory inputs, not solely depending on auditory input channels.

15) Related to the previous point, the authors focus on the auditory processing in this pathway, but almost all of their manipulations are performed during training and affect shock processing as well. In the one experiment they perform to inhibit CS evoked activity during the probe test (Fig. 1F-G) they see

a minimal effect of the manipulation on freezing. An alternate possibility is that this pathway is important for conveying aversive shock information to the lateral amygdala. They could address this by looking more carefully at the shock responses in the striatal neurons and in the SNc inputs/dopamine responses (as suggested above), but my guess is that they will see shock responses throughout the system. Ideally, experiments using more temporally specific optogenetic inhibition during the shock period of training would be performed. Without this they should acknowledge this possibility in the Discussion.

This is also a great point. We performed the suggested analyses and added a brief discussion on alternative interpretation.

We aligned neuronal ROI activities in the auditory striatum and SNc to the onset of foot shocks, to explore potential shock responses from these neurons. As we presented in response to #4, we found that there are auditory striatal neurons responding to the foot shocks (52 out of 262. **Supple. Fig. 4B**), and they are not aligned to those tone responsive neurons (188 out of 262). Interestingly, 25 out of 58 recorded SNc neurons showed responses to foot shocks, and not aligned to those tone responsive neurons either (23 out of 58). These data suggest that indeed both the auditory striatal neurons and the striatal-projecting SNc neurons carry somatosensory aversive information, which is likely independent from the auditory information. We agree with the reviewer that this is important to know and we included this information in the revised **Suppl. Fig. 5H**.

As the reviewer had pointed out, our optogenetic studies did not separate the roles of auditory processing and shock processing in the auditory striatum and SNc. Thus, it is possible that somatosensory information expression in the auditory striatum and SNc could also contribute to fear memory processing and that this information could be relayed to the amygdala as well. Based on this, we add the following to the discussion on page 19:

“Related to our study, we primarily tested the necessity of auditory processing in the auditory striatum and auditory striatal-projecting SNc neurons, but did not assess its sufficiency nor the role of shock expression. Indeed, we found that neurons in the auditory striatum and the SNc exhibited shock responsivity (**Suppl. Figs. 4B&5H**) and thus this aversive somatosensory processing could also contribute to fear memory and modulation of the amygdala. Somatosensory processing has been implicated in the tail striatum and thus future studies could be conducted to investigate whether there

is an integration of auditory and somatosensory cues within these pathways (Hong et al., 2018; Takahashi et al., 2020).”

Supplementary Figure 4

R1Q15 (Suppl. Figs. 4B&5H). B, Left, heatmap of averaged $\Delta F/F$ auditory striatal neuronal responses towards foot shock, arranged by descending order of peak amplitude. Right, the same neuronal ROIs arranged by tonal response amplitude with extended time course to include responses to foot shocks starting at the 20 s time point. $n = 262$ neuronal ROIs from 6 mice.

Supplementary Figure 5

H, Left, heatmap of averaged $\Delta F/F$ SNc neuronal responses towards foot shock arranged by descending order of peak amplitude. Right, the same neuronal ROIs arranged by tonal response amplitude with extended time course to include responses to foot shocks starting at the 20 s time point. $n=58$ neuronal ROIs from 6 mice.

16) *The mechanism through which the striatal neurons regulate the lateral amygdala during learning is not clear and the authors should discuss this at more length. It seems they want to say that the striatal neurons projecting to amygdala transmit associative information about the auditory CS which boosts, in some way, lateral amygdala processing. If so, one question is how GABAergic (medium spiny) striatal neurons this do when the lateral amygdala neurons are known to develop excitation to the CS during fear conditioning (1). Thus, it is not clear how this inhibitory striatal signal could facilitate lateral amygdala excitatory associative processing. Of course, there are inhibitory networks in lateral amygdala which could be used for disinhibition and some recent studies report reductions in CS-evoked responding following conditioning, but the authors should address this point more clearly.*

We appreciate the reviewer's insight and suggestion regarding how the auditory striatal neurons may regulate the LA fear circuit to impact fear memory. Indeed, we think it is possible that the auditory striatum provides a disinhibition drive to the LA output by targeting the interneurons in LA. This can be a very exciting hypothesis to start off to determine how LA relays the striatal inputs. To incorporate the suggestions, we revised our discussion on pages 21-22, also copied below.

“On a circuit mechanism level, the auditory striatum like other striatal regions, is comprised of solely GABAergic neuronal populations, and thus may provide a disinhibitory drive to gate learning in principle amygdalar neurons, akin to local interneuron population function (d'Aquin et al., 2022; Krabbe et al., 2019; Lucas, Jegarl, Morishita, & Clem, 2016; Morrison et al., 2016; Stujenske et al., 2022; Wolff et al., 2014). In this context, evidence indicates that the parvalbumin and somatostatin neuronal populations of the lateral amygdala and basolateral amygdala tightly controls the output and plasticity of the amygdala for fear acquisition, discrimination, as well as for extinction (e.g. for fear extinction, the gradual interneuron inhibition of amygdalar output as a mechanism for fear memory extinction and dampening) (Davis, Zaki, Maguire, & Reijmers, 2017; Krabbe, Grundemann, & Luthi, 2018; Lucas et al., 2016; Morrison et al., 2016; Polepalli, Sullivan, Yanagawa, & Sah, 2010; Stujenske et al., 2022). One possibility is that the auditory striatum could provide inhibition to these local interneurons to disinhibit and enhance lateral amygdala principle neuron activity – thus the auditory striatum may serve as a parallel reinforcement circuit to fine tune the sensory learning functions of the lateral amygdala. Consistent with this, in the lateral amygdala, PV neuronal activity has been found to be decreased throughout fear memory formation (Lucas et al., 2016). The auditory striatum itself has been found to be involved in operant and reward-based learning and could provide a source of disinhibitory plasticity in a variety of contexts beyond fear learning (L. Chen et al., 2019; Grosso, Santoni, Manassero, Renna, & Sacchetti, 2018; Krabbe et al., 2019; Letzkus, Wolff, & Luthi, 2015; Wolff et al., 2014; Xiong et al., 2015). Thus, future work will be required to investigate the different possible valences that the auditory striatum could relay to the amygdala. Furthermore, it will be interesting and important to determine whether auditory striatal input impacts the plasticity of either auditory cortical or auditory thalamic inputs to the lateral amygdala, projections found to be critical for auditory fear memory formation (Herry & Johansen, 2014; Romanski & LeDoux, 1992).”

Minor:

1) *Regarding freezing categorization, more information should be provided. Does the software differentiate between immobility (e.g., grooming) and freezing? Additionally, do all freezing quantifications cover the entire session duration (including CS+ intertrial periods), or are they only taken during CS presentation?*

We thank the reviewer for pointing out this missing detailed information. We revised the corresponding method section to include these details (page 27). “This threshold of quantification distinguishes freezing behavior from generalized pause-groom behavior. Freezing is quantified as a function of time throughout the entire behavioral session (habituation, conditioning, probe). Overall quantification of freezing percentage results from averaging of freezing percentage at fixed time intervals.”

2) *Fig1. B Legend: “Bilateral implantation and infusion of AAV-Cre with either AAV-DIO-GFP or AAV-DIO-ArchT”. The schema suggests a virus injection of AAV-GFP and AAV-ArchT.*

We have corrected the Figure legend.

3) *The calcium imaging activity graphs focus solely on neuron CS representation. Including information on neuron responses to the US during conditioning could provide additional valuable insights into how TS and DA neurons of the SNc encode sensory aversive information.*

We performed and added the suggested analysis (**Suppl. Figs. 4B&5H**) and discussion (page 19), similar to the related responses to #4 and #15.

4) *Fig4. B and F, Calcium imaging graph should have the same scale and same sizing to facilitate lecture and comparison.*

We have revised the graphs accordingly.

5) *Supplementary Fig.2: Histology picture of viral ablation of the auditory striatal of a representative animal should be appreciable.*

We have included representative images for both global ablation and projection-specific ablation in the auditory striatum in the **Suppl. Fig. 2D&F**.

6) *Supplementary Fig.2: In the Fig.2 legend "Shade of red in intensities represent..." Must be corrected by blue.*

We have revised the legend accordingly.

Reviewer #2:

In this manuscript, Chen et al. demonstrate the role of the auditory striatum in classical auditory fear conditioning behavior in mice. The authors use a combination of circuit-specific opto- and chemogenetics as well as in vivo fluorescence imaging in freely behaving mice to show that the auditory striatum is necessary for proper fear memory formation. Mice show reduced freezing during future tone presentations when the auditory striatum was inhibited during conditioning and neuronal activity in the striatum is potentiated following fear conditioning. The authors further show that dopamine signaling in the auditory striatum is also significantly modulated following fear conditioning and might be responsible for striatal plasticity. This study finds that the role of the auditory striatum in fear memory is likely mediated by its projections to the lateral amygdala.

Overall, this manuscript highlights a novel and important behavioral role for the auditory striatum, a thus far understudied region of the brain. The experiments in this study are well-designed and the authors included many important control experiments. However, a couple of the findings warrant some additional experiments and/or analysis and discussion to substantiate the authors' claims.

1. One curious aspect of the findings is that auditory striatum or dopaminergic input to the auditory striatum during conditioning does not affect freezing during the conditioning session, but only subsequent probe sessions. Potentiation of calcium signals in auditory striatum neurons, however, is observed during conditioning. How do the authors think about this? Does fear conditioning-induced striatal potentiation during the conditioning session also get abolished with chemogenetic inhibition of SNc inputs (Fig.5)?

In response to the reviewer's comment, we performed and included new analyses showing that chemogenetic inhibition of SNc inputs abolished conditioning-induced striatal neuronal potentiation (**Fig. 5F**). We agree with the reviewer that it is interesting to observe no effect on freezing during conditioning when we inhibit the striatal or SNc activities. We have also been puzzled by this. One possibility is that the freezing behavior in conditioning session, induced by the foot shock from the somatosensory circuitry, is acute and strong and masks the additional changes from the auditory circuitry. In addition, the potentiation of striatal neuronal activity during conditioning may be the plasticity involved in the modulation of amygdala for memory consolidation rather than support of acute freezing behavior. To include the understanding of this interesting point, we added a paragraph in the discussion on this (pages 16-17). The discussion is copied below.

"An interesting observation is that while we observed enhancement of striatal neural activity as well as dopaminergic transmission during the conditioning, inhibition of these circuits did not appear to impact the development of freezing behavior in conditioning sessions but only affect freezing responses in probe sessions. One possible explanation is that the freezing behaviors to the US (foot shock) during conditioning is through the somatosensory circuitry (Wang et al, 2021; Gross et al., 2012), which is acute and strong enough to mask the additional impact from the auditory circuitry. It is conceivable that in the context of the behavior the amygdala could employ other parallel pathways to enforce freezing behavior acutely (Kim et al, 2013; Jhang et al., 2018). Along with this, the potentiation of striatal neuronal activity during conditioning may be the plasticity involved in modulation of amygdala over a longer time scale, akin to memory consolidation."

2. Looking at the degree of ablation in the caspase experiments (Supp. Fig. 2C) suggests that over 75% of neurons in the auditory striatum are projecting to the lateral amygdala (and thus receiving cre from CAV-cre). This seems rather high, and it might be necessary to add a control experiment showing the specificity of DIO-Caspase (AAV-DIO-Caspase injection without CAV-cre).

We thank the reviewer for pointing out this confusion, which was from the mistake that we mislabeled the unrestricted ablation results. In the revision, we have corrected the mistake by providing representative images, quantifications, and corresponding behavioral results for both unrestricted ablation and projection-specific ablation in the auditory striatum. We included these results in **Suppl. Fig. 2** and have modified the text accordingly on pages 8-9.

Unrestricted ablation of neurons in the auditory striatum was performed by injecting AAV-Caspase into the auditory striatum (**Suppl. Fig. 2B**). Projection-specific ablation of auditory striatal neurons was performed by injecting CAV-Cre into lateral amygdala and AAV-DIO-Caspase into the auditory striatum (**Fig. 2C**). Both ablation strategies led to reduced freezing behaviors in the auditory fear conditioning (**Fig. 2C & Suppl. Fig. 2C**).

In unrestricted ablation, there is a large reduction of neurons within the auditory striatum, while sparing those in the lateral amygdala and neighboring regions (**Suppl. Fig. 2D&E**). The projection-specific ablation caused less reduction of auditory striatal neurons (**Suppl. Fig. 2F&G**, 32% reduction).

R2Q2 (Suppl. Fig. 2B-G) Caspase-mediated ablation of the auditory striatal neurons. B. Schematic for ablation of the auditory striatum without genetic restriction. **C.** Freezing behavior across cohorts of control vs. ablation. Error bars are SEM (unpaired Mann-Whitney test; ns $p > 0.05$, * $p < 0.05$; ** $p < 0.01$). **D&E**, unrestricted ablation. **F&G**, projection-specific ablation. **D.** Images, auditory striatum in control animal with lateral amygdala shown, and comparable image with caspase ablation. The bottom two panels are respective zoomed-in images. Right graph, Quantification of neuronal density in the auditory striatum between control and caspase groups for unrestricted ablation Control, 1.10 ± 0.02 ; Ablation, 0.20 ± 0.03 neuron $\times 10^3/\text{mm}^3$. Error bars are SEM (unpaired Mann-Whitney test; **** $p < 0.0001$). **E.** Schematic of viral spread and ablation spread for both control and ablation groups for experimental data shown in **D**. **F.** Histology of amygdala-projecting auditory striatum neuronal ablation experiment performed in **Fig. 2**. Images, auditory striatum in control animal with lateral amygdala shown, and comparable image with projection-specific caspase ablation. The bottom two panels are respective zoomed-in images. Graph, quantification of neuronal density in the auditory striatum between control and caspase groups for projection-specific ablation. Control, 1.07 ± 0.04 ; Ablation, 0.72 ± 0.08 neuron $\times 10^3/\text{mm}^3$. Error bars are SEM (unpaired Mann-Whitney test; **** $p < 0.0001$). **G.** Schematic of viral spread

spread and ablation spread for both control and ablation groups for experimental data shown in **F**. For histological images, labeled in blue is DAPI and green is the neuronal marker NeuN. Scale bars are 200 μm and 100 μm for overall and zoomed-in images, respectively.

3. Given the large percentage of neurons in the auditory striatum that is affected by viral Caspase ablation, it is difficult to conclude from this experiment that an auditory striatal projection to the lateral amygdala is the main circuit mediating the striatal role in auditory fear conditioning. Optogenetic inhibition/stimulation of projections from the auditory striatum in the amygdala would be more specific. Given the unusual direct projection from the striatum to the amygdala, this circuit warrants further characterization. For example, are D1 or D2 expressing neurons projecting to the amygdala? Are these just minor collaterals of more traditional striatal projection targets or is the amygdala indeed the main output of the auditory striatum? Please at least discuss.

We thank the reviewer for pointing out this issue. In the revised manuscript and below response, we clarified the confusion on Caspase ablation results; provided more data and added a discussion (page 17) to address this concern.

In **Fig. 2C&D** and **Suppl. Fig. 2B-G**, we presented both global ablation and projection-specific ablation results. While injection of nonrestricted AAV expressing Caspase in the auditory striatum caused about 80% reduction of neurons (**Suppl. Fig. 2B&D**), injection of Cre dependent AAV-DIO-Caspase in the auditory striatum with CAV-Cre injection in the lateral amygdala (**Fig. 2C**) caused about 30% reduction of neurons (**Suppl. Fig. 2F**). However, both approaches induced similar levels of fear memory deficits (**Fig. 2D & Suppl. Fig. 2C**). Thus, we believe the amygdala is the major target of the auditory striatum in fear conditioning. Nonetheless, we agree with the reviewer that optogenetic manipulations of auditory striatal terminals in the lateral amygdala would further confirm the pathway's specificity. We included it in the Discussion section. Also copied below.

We also agree with the reviewer that the original results do not differentiate the activities of the two major populations (D1 and D2 MSNs) in the auditory striatum. We performed anterograde viral tracing experiments and found that both D1 and D2 MSNs project to the lateral amygdala. In brief, we injected AAV-DIO-tdTomato into the auditory striatum of D1-Cre and D2-Cre (A2a-Cre) mice. Infected neurons were found in the auditory striatum identified as D1-MSNs or D1-MSNs, respectively. The tdTomato+ axon terminals (pseudo-colored in green) were found in the lateral amygdala of both cohorts of mice. Further studies will need to be done to examine whether they exhibit differential activity patterns in tone-cued fear conditioning. We added this limitation in the **Discussion** section. Also copied below.

“Compared to the more traditional striatal targets such as the globus pallidus, the amygdala receives much less attention. In this study we demonstrated a direct connection from the auditory striatum to the lateral amygdala (Fig. 2 & Supple. Fig. 2). Ablation of amygdala-projecting neurons in the auditory striatum dramatically impaired fear memory. However, our results did not rule out possible participation of collaterals from these neurons to other brain regions. Optogenetic manipulations of striatal terminals in the lateral amygdala would confirm this pathway's specificity. Furthermore, our results do not differentiate the activities of striatal neuronal populations (e.g. D1 and D2 MSNs). Anterograde viral tracing suggests that both D1 and D2 MSNs project to the lateral amygdala. How dopaminergic modulation through these neurons may impact auditory fear conditioning? In the literature, both D1 and D2 receptors systemic pharmacological antagonism has been shown to abrogate different forms of fear conditioning with mixed interpretations (De Bundel et al., 2016; de Souza Caetano et al., 2013; Greba & Kokkinidis, 2000; Guarraci et al., 1999; Ikegami et al., 2014; Inoue et al., 2000; Ponnusamy et al., 2005). Differential roles for the D1 and D2 MSN pathways within the dorsal and ventral striatal subdivisions have been suggested that the pathways may correspond to appetitive-positive-actions or aversive-negative-inhibitions depending on the behavior being studied (Gerfen, 2022; Isett et al., 2023; Nishioka et al., 2023). However, this D1 and D2 opponency is nuanced, with evidence suggesting that the D1 pathway supports generalized learning while the D2 pathway could aid in refinement of learning (Iino et al., 2020; Matamales et al., 2020). In the auditory striatum, both D1 and D2 MSNs are responsive towards auditory cues (Nardoci et al., 2022). Our previous works indicate that the D1

pathway is important for auditory discrimination and task performance, while the D2 pathway appears to be dispensable (Chen et al., 2022). Future studies are needed to examine the potential functional differences of D1 and D2 MSNs in fear conditioning.”

R2Q3. Both D1 and D2 MSNs in the auditory striatum project to the lateral amygdala. **A**, histology of anterograde tracing from the auditory striatum in a D1-Cre mouse with Cre-dependent AAV-tdTomato (pseudo-color green). Left, representative image of the injection site. Right, zoomed-in images of infected cell bodies in the auditory striatum and axon terminals in the lateral amygdala. **B**, same as in A but from a D2-Cre (A2a-Cre) mouse. Scale bars for large-view images and zoomed-in images are 200 μm and 20 μm , respectively.

4. For statistical quantification of freezing in Fig. 1 E and 1G the authors seem to use the response to individual tones as individual data points, however, the authors should average the response for each mouse and treat individual mice as data points for analysis since multiple responses to tones in the same mouse are not independent observations. A similar issue seems to exist for calcium responses in Fig. 4D and H.

We thank the reviewer for pointing out this issue. In the revised **Fig. 1E&G** and **Fig. 5C&G**, we now used the averaged freezing responses from individual mice as data points for statistical quantifications. In **Fig. 4H**, each data point represents the whole field DA sensor signal changes from each mouse. In **Fig. 4D**, to also include the information of heterogeneity of the SNc neuronal activity, each data point represents each recorded SNc neuron.

5. In general, it is not clear what individual data points plotted in many figures represent. For example, in Fig. 4C and D it is suggested that each circle is an individual neuronal ROI, and in Fig 4 G and H in each point should be an individual GRIN FOV. However, the number of circles does not match the n number stated in the legends.

We thank the reviewer for pointing out the issue across figures. We now revisited all the dataset related to all the figures and have revised all figures and legends accordingly to clarify each data point's representation.

6. The authors include a great control experiment testing for calcium responses to a tone not paired with shocks (Suppl. Fig. 3A-C), however, it would be important to also show the calcium response to the paired tone (Tone A) for this set of mice.

We agree with the reviewer and have included the calcium response to the paired tone (Tone A) in the revised **Suppl. Fig. 3B&C**. (Also shown below)

R2Q6 (Suppl. Fig. 3A-E). Auditory striatal neurons show potentiated responses to paired tones but not non-paired tones. **A**, Schematic for fear conditioning with Tone A paired with a shock and an innocuous Tone B. Eight 20-s, 10-kHz tones (Tone B) were presented to mice during a pre-conditioning session in a similar manner as in habituation. Subsequently, mice underwent auditory fear conditioning as in Fig. 1 with 20-s Tone A. One day later, mice were again presented with Tone B in a different context. **B**, Heatmap showing neuronal ROI responses to Tone A (left two plots) and Tone B (right two plots) in pre- and post-conditioning sessions. **C**, Averaged neuronal activity in response to Tone A (left) and Tone B (right) during pre- and post-conditioning sessions. Error bars are SEM (n = 114 neuronal ROI across 6 mice; Wilcoxon rank-sum test; ns, p > 0.05). **D**, Corresponding behavioral data for striatal neuronal imaging analysis. Left, averaged freezing percentage in response to tone A during habituation, conditioning, and probe sessions. Right, averaged freezing in response to tone B during pre-conditioning and post-conditioning sessions (3 post-conditioning tone B sessions; n = 4 mice). **E**, Left, example histology of lens placement and GCaMP6f expressed in the auditory striatum. Green, GCaMP6f; purple, DAPI. Scale bar = 1.0 mm. Right, summary of lens implantation sites. Each green line represents the bottom tip of a lens for each mouse.

7. Authors should show the freezing behavior for mice in which calcium imaging in the auditory striatum was performed (Fig. 3 and 5E-G), similar to what was done for SNc recordings.

We thank the reviewer for pointing out this missing information. In the revised **Fig. 5G** and **Suppl. Fig. 3G**, we included freezing behavior quantifications from mice that were used for chemogenetic manipulation and calcium imaging, respectively.

8. What are the criteria used to determine which of the detected ROIs to include in the calcium image analysis? E.g., Fig.4D says that only 23 out of 58 detected neuronal ROIs are plotted. It was indeed unclear in the first submission. We added clarifications in the result (page 13) and Method (page 33) sections. The ROIs/neurons presented are those that demonstrated tone responsiveness during any one of the sessions plotted. We clarify this in the methods as well as the corresponding parts of the manuscript.

Minor points

9. Text in results mentions that freezing was $12.32 \pm 1.02\%$ during habituation, however in Fig. 1C it looks like the average should be below 10%.

We thank the reviewer for pointing out this mistake. We corrected the quantification in the results section on page 7: $7.28 \pm 0.89\%$.

10. The results state that all tones are interspersed with 100- to 180-s intertrial intervals, but the methods state 60-100 s intertrial intervals.

This was a mistake. We corrected the method section to reflect the experimentation which was conducted with a 100- to 180-s intertrial interval (page 26).

11. Please state the diameter of optic fiber cannula and GRIN lens implants used.

We added this missing information in the method section (pages 25-26): The optic fibers used for this study were 0.2 mm in diameter. The lenses used for this study were either 0.6 mm or 0.5 mm in diameter.

12. In some figures the numbers of mice/neurons are missing (e.g., Supp. Fig. 1G, Supp. Fig. 3). We added the missing information to all figure legends.

Reviewer #3

In this study Chen and colleagues explored the function of the auditory striatum (or the “tail of the striatum”) in auditory fear conditioning. They report that optogenetically inhibiting auditory striatal neurons impaired fear memory formation, which was mediated through the striatal-lateral amygdala pathway. Furthermore, they used calcium imaging in behaving mice to show that auditory striatal neuronal responses to conditioned tones were potentiated across memory acquisition and expression. They also provided evidence suggesting that nigrostriatal dopaminergic projections may play a role in modulating conditioning-induced striatal potentiation. This study used a combination of cutting-edge technologies to convincingly show the plasticity of auditory striatum neuron response plasticity and its involvement in fear conditioning. In addition, the study provides nice evidence suggesting the involvement of a nigro-striatal-amygdala circuit in auditory fear conditioning. These findings are novel and further our understanding of the neural circuits underlying fear conditioning. In general, I think this paper fits nicely to Nature Communications. I have some suggestions that may strengthen the major conclusions.

1. The authors propose a nigro-striatal-amygdala circuit for auditory-conditioned fear memory formation and expression. It would be nice to assess the flow of information in this circuit in their imaging data. For example, do dopamine neurons fire earlier than neurons in auditory striatum in response to CS during conditioning or memory retrieval? From the traces in Figure 4B and those in Figure 3E, it seems that there is a longer latency for dopamine neurons. However, this could be skewed by averaging data, so a cell-by-cell analysis on response latency is needed for each of these two populations. It would be interesting to see if the distributions of response latencies are different or not.

2. Related to #1 above, it seems that the dopamine neuron responses (Figure 4B) had a longer latency than the dopamine sensor responses (Figure 4F). Again, a cell-by-cell analysis on dopamine neuron response latency is needed, to see whether some dopamine cells had short latencies.

We thank the reviewer for these deep insights and pointing out the confusion. Since they are related, we combined our responses to these two points (#1 & #2) below. In response to the suggestion and confusion, we performed new analyses and revised the figures. In brief, we updated representative traces and populational analyses to address this concern (**Fig. 4B&C** and **Suppl. Fig. 5G**) and added a discussion to include the limitations and possibilities (pages 20-21).

We included a heatmap to demonstrate individual tone responsive SNc neuronal responses (**Fig. 4C** or see below **R3Q1&2**). Most SNc neuronal responses exhibit shorter latency than the striatal neuronal responses (**Suppl. Fig. 5G** or see below **R3Q1&2**). However, there are a few SNc neurons that displayed substantially long latency. To represent the majority of SNc neuronal responses, we replaced the **Fig. 4B** example traces with the ones of shorter latency.

Again, the reviewer raised an interesting hypothesis regarding the activity flow. We investigated individual responses in the SNc (GCaMP6f) and auditory striatum (DA2m). The variation turned out to be big. We then investigated some analysis of dynamics with these two genetic sensors. The kinetics and peak of fluorescence between GCaMP6f (SNc neuronal activity) and DA2m (DA activity) are different and the onset of rise in fluorescence are good for activity analyses but seems insufficient for quantitative analyses of activity onset. For example, real time calcium fluctuation may be sufficient to trigger DA release and such a change may not be reflected by *in vivo* imaging. With these two sensors, we found we do not have a good temporal resolution for this activity flow analysis. As the reviewer mentioned, it is possible that VGLUT2 activity may locally trigger dopaminergic release from axon terminals. Another possibility could be local cholinergic activation. These are very interesting possibilities to be tested out for our future work. Due to these interesting insights, we added a discussion to include the limitations and possibilities (pages 20-21).

Figure 4

R3Q1&2 (Fig. 4C & Suppl. Fig. 5G). C, Averaged tonal response latencies for individual SNc neuronal ROI (n = 23 neuronal ROI from 6 mice). **G**. Comparison of tonal response latencies from SNc neuronal ROI, DA sensor FOV (n = 6 mice), and auditory striatal neuronal ROI (n = 188 from 6 mice). Error bars are SEM (Mann-Whitney test; ns, $p > 0.05$; ** $p < 0.01$).

Supplementary Figure 5

3. The authors need to provide evidence that the retrograde strategy shown in Figure 4A and Figure 5E actually targeted dopamine neurons in the SNc.

We thank the reviewer for pointing out this issue. We performed immunohistochemistry staining to confirm these neurons' identity, presented a new figure, revised our statement, and discussed the limitation.

We injected the retrograde labeling virus CAV expressing mCherry (CAV-mCherry) into the auditory striatum and identified the mCherry+ neurons in the SNc (**R3Q3A**). We then performed TH staining to identify dopaminergic neurons in the SNc and quantified the numbers of TH+ neurons and mCherry+ neurons. We found that $94.8 \pm 1.92\%$ mCherry+ neurons are TH+ and $5.4 \pm 2.07\%$ neurons are TH- (**R3Q3B**), consistent with our previous published work (**Fig. 1**, (Chen et al., 2022)). Thus, we conclude that the majority of SNc neurons projecting to the auditory striatum are dopaminergic. We included this new analysis in **Suppl. Fig. 5A&B**. Accordingly, we revised the text on pages 12-13. We agree that the labeling strategy in this recording would not preclude the small portion of non-dopaminergic neurons (~5% according to the above tracing quantification). It will be interesting to dissect their activity during the task, with which we unfortunately do not have an in-house method. Therefore, we added a short discussion on this limitation (page 20).

R3Q3 (Suppl. Fig. 5A&B). Retrograde tracing from the auditory striatum primarily labels dopaminergic neurons in the substantia nigra pars compacta (SNc). **A.** Left upper, injection schematic of CAV-mCherry into the auditory striatum. Left bottom, representative imaging of injection site and traversing mCherry+ axons in the auditory striatum. Right, retrograde labeling in midbrain dopamine regions, the VTA and SNc. mCherry is pseudo-colored in gold, with TH labeling in blue. Scale bar is 100 μ m. **B.** The left two bars show percentage quantification of mCherry+ neurons in the VTA and SNc. The right two bars show the percentage of mCherry+ neurons that are either TH- or TH+ neurons.

4. Supplementary Figure 2: please provide representative histology images to show the viral ablation effect.

We thank the reviewer for pointing out this missing information. In response to this concern and a concern from the Review 1 (#2), we provided representative images and performed additional quantifications and corresponding behavioral results for both unrestricted ablation and projection-specific ablation in the auditory striatum. We included these results in **Suppl. Fig. 2** and have modified the text accordingly on pages 8-9.

Unrestricted ablation of neurons in the auditory striatum was performed by injecting AAV-Caspase into the auditory striatum (**Suppl. Fig. 2B**). Projection-specific ablation of auditory striatal neurons was performed by injecting CAV-Cre into lateral amygdala and AAV-DIO-Caspase into the

auditory striatum (**Fig. 2C**). Both ablation strategies led to reduced freezing behaviors in the auditory fear conditioning (**Fig. 2C & Suppl. Fig. 2C**).

In unrestricted ablation, there is a large reduction of neurons within the auditory striatum, while sparing those in the lateral amygdala and neighboring regions (**Suppl. Fig. 2D&E**). The projection-specific ablation caused less reduction of auditory striatal neurons (**Suppl. Fig. 2F&G**, 32% reduction).

R3Q4 (Suppl. Fig. 2B-G) Caspase-mediated ablation of the auditory striatal neurons. B. Schematic for ablation of the auditory striatum without genetic restriction. **C.** Freezing behavior across cohorts of control vs. ablation. Error bars are SEM (unpaired Mann-Whitney test; ns p > 0.05, *p < 0.05; **p < 0.01). **D&E**, unrestricted ablation. **F&G**, projection-specific ablation. **D.** Images, auditory striatum in control animal with lateral amygdala shown, and comparable image with caspase ablation. The bottom two panels are respective zoomed-in images. Right graph, Quantification of neuronal density in the auditory striatum between control and caspase groups for unrestricted ablation Control, 1.10 ± 0.02 ; Ablation, 0.20 ± 0.03 neuron $\times 10^3/\text{mm}^3$. Error bars are SEM (unpaired Mann-Whitney test; ****p < 0.0001). **E.** Schematic of viral spread and ablation spread for both control and ablation groups for experimental data shown in **D**. **F.** Histology of amygdala-projecting auditory striatum neuronal ablation experiment performed in **Fig. 2**. Images, auditory striatum in control animal with lateral amygdala shown, and comparable image with projection-specific caspase ablation. The bottom two panels are respective zoomed-in images. Graph, quantification of neuronal density in the auditory striatum between control and caspase groups for projection-specific ablation. Control, 1.07 ± 0.04 ; Ablation, 0.72 ± 0.08 neuron $\times 10^3/\text{mm}^3$. Error bars are SEM (unpaired Mann-Whitney test; ****p < 0.0001). **G.** Schematic of viral spread and ablation spread for both control and ablation groups for experimental data shown in **F**. For histological images, labeled in blue is DAPI and green is the neuronal marker NeuN. Scale bars are 200 μm and 100 μm for overall and zoomed-in images, respectively.

5. Page 15, last paragraph: avoid using “imaging” to describe photometry data, as photometry is not imaging.

We thank the reviewer for pointing out this confusion. Because we used whole field-of-view analysis and do not focus on spatial quantification, microendoscopic method is indeed similar to the use of fiber photometry to capture DA2m signals. To clarify this confusion, we’ve added a sentence in the result section (page 14): “We note here that our methodology and analysis is functionally similar to prior use of fiber photometry to capture DA2m signals (Sun et al., 2020).”

REVIEWER COMMENTS

Reviewer #1 (Remarks to the Author):

The authors have addressed most of my previous concerns. However, one issue remains. Related to my previous comment 1 regarding potential anterograde/retrograde viral spread in Fig. 2A and B, the authors now show that anterograde viral spread into amygdala is minimal. However, they did not show representative images or discuss potential spread from lateral amygdala into the auditory striatum for the retrograde tracing experiments (Fig. 2B). This may be difficult because it might be hard to distinguish between local infectivity and retrogradely labeled cells, particularly because of the proximity of the LA to the auditory striatum. However, this is important as it does appear that in the anterograde experiments, there was spread outside of the auditory striatum (though not to lateral amygdala). One thing the authors could do is inject a similar AAV (same serotype, fluorophore and concentration for example) into the lateral amygdala which is not retrograde at the volumes they used in their retrograde tracing experiments to test the spread. Alternatively, they may be able to expand on their data using the retrograde (from lateral amygdala) caspase cell killing dataset which seems to show that this approach only kills cells in auditory striatum and not lateral amygdala if the injection volumes/retrograde viruses are similar.

Reviewer #2 (Remarks to the Author):

The authors have done an excellent job addressing all the major and minor concerns. The manuscript has been significantly improved. Congrats on an innovative and exciting work!

Reviewer #3 (Remarks to the Author):

The authors have thoroughly addressed all my questions. Overall, the revised paper is much improved. I have no further concerns.

We thank the reviewers for their remarks and comments on our revision. We have taken the suggestions and revised the manuscript further according to the remaining concern on the viral spread issue. (reviewer's comments in grey and italic, our responses in black).

Reviewer #1 (Remarks to the Author):

The authors have addressed most of my previous concerns. However, one issue remains. Related to my previous comment 1 regarding potential anterograde/retrograde viral spread in Fig. 2A and B, the authors now show that anterograde viral spread into amygdala is minimal. However, they did not show representative images or discuss potential spread from lateral amygdala into the auditory striatum for the retrograde tracing experiments (Fig. 2B). This may be difficult because it might be hard to distinguish between local infectivity and retrogradely labeled cells, particularly because of the proximity of the LA to the auditory striatum. However, this is important as it does appear that in the anterograde experiments, there was spread outside of the auditory striatum (though not to lateral amygdala). One thing the authors could do is inject a similar AAV (same serotype, fluorophore and concentration for example) into the lateral amygdala which is not retrograde at the volumes they used in their retrograde tracing experiments to test the spread. Alternatively, they may be able to expand on their data using the retrograde (from lateral amygdala) caspase cell killing dataset which seems to show that this approach only kills cells in auditory striatum and not lateral amygdala if the injection volumes/retrograde viruses are similar.

We thank the reviewer for pointing out this remaining concern on the viral spread. Following the reviewer's suggestion, we presented below images of differing levels of spread in terms of striatal neuronal ablation in the CAV-Cre/caspase experiments (**Response Fig. 1A**), and quantified the neuronal density in the lateral amygdala for both ablation strategies (**Response Fig. 1B**). This new analysis is included in the revised **Supplementary Figure 2 D&F**, and the corresponding text on page 9.

In **Response Fig. 1A** we presented 3 example images from 3 different mice to demonstrate the striatal ablation, from largest to smallest, in the CAV-Cre/caspase experiments presented in the manuscript **Figure 2C**. In each of the animals, it is apparent that there is significant reduction in auditory striatum neuronal density, but no appreciable spread towards the lateral amygdala. We then performed tdT NeuN density analysis for the lateral amygdala region across animals presented in both the straightforward ablation as well as retrograde specific ablation. As shown in **Response Fig. 1B**, there was limited spread of the caspase virus to the lateral amygdala, resulting in similar neuronal densities in the lateral amygdala across control and viral ablation cohorts.

Response Figure 1. A, Histology of amygdala-projecting auditory striatum neuronal ablation experiment performed in **Figure 2**. 3 example images from 3 different mice show varying degrees of ablation from largest to smallest. **B**, Quantification of neuronal density in the lateral amygdala between control and caspase groups for both straight-forward (left) and projection-specific ablation (right). Straight forward

ablation: Control, 2.45 ± 0.09 ; Ablation, 2.31 ± 0.10 neuron $\times 10^3/\text{mm}^3$. Projection-specific ablation: Control 2.22 ± 0.18 neuron $\times 10^3/\text{mm}^3$; Ablation, 2.45 ± 0.11 neuron $\times 10^3/\text{mm}^3$. Error bars are SEM (unpaired Mann-Whitney test; ns $p > 0.05$).

REVIEWERS' COMMENTS

Reviewer #1 (Remarks to the Author):

The authors have addressed my remaining concern and I support publication of the revised manuscript.